# Next-generation large-scale binary protein interaction network for *Drosophila melanogaster*

Hong-Wen Tang [1,2,3,16], Kerstin Spirohn [1,4,16], Yanhui Hu[1], Tong Hao [1,4], István A. Kovács [4,5,6], Yue Gao[1], Richard Binari[1,7], Donghui Yang-Zhou[1], Kenneth H. Wan [8], Joel S. Bader [9,10], Dawit Balcha[1,4], Wenting Bian[1,4], Benjamin W. Booth[8], Atina G. Coté[4,11,12], Steffi de Rouck[13], Alice Desbuleux[1,4], Kah Yong Goh[2], Dae-Kyum Kim[14], Jennifer J. Knapp [11,12], Wen Xing Lee[2], Irma Lemmens[13], Cathleen Li[1], Mian Li[1], Roujia Li[4,11,12], Hyobin Julianne Lim[14], Yifang Liu[1], Katja Luck [1,4], Dylan Markey[1,4], Carl Pollis [1,4], Sudharshan Rangarajan[1,4], Jonathan Rodiger [1], Sadie Schlabach[1,4], Yun Shen[1,4], Dayag Sheykhkarimli [4,11,12], Bridget TeeKing[1,4], Frederick P. Roth [4,11,12,15], Jan Tavernier[13], Michael A. Calderwood [1,4], David E. Hill [1,4], Susan E. Celniker[8] ✉, Marc Vidal[1,4] ✉, Norbert Perrimon [1,7] ✉ & Stephanie E. Mohr [1] ✉

Generating reference maps of interactome networks illuminates genetic studies by providing a protein-centric approach to finding new components of existing pathways, complexes, and processes. We apply state-of-the-art methods to identify binary protein-protein interactions (PPIs) for *Drosophila melanogaster*. Four all-by-all yeast two-hybrid (Y2H) screens of > 10,000 *Drosophila* proteins result in the 'FlyBi' dataset of 8723 PPIs among 2939 proteins. Testing subsets of data from FlyBi and previous PPI studies using an orthogonal assay allows for normalization of data quality; subsequent integration of FlyBi and previous data results in an expanded binary *Drosophila* reference interaction network, DroRI, comprising 17,232 interactions among 6511 proteins. We use FlyBi data to generate an autophagy network, then validate in vivo using autophagy-related assays. The *deformed wings* (*dwg*) gene encodes a protein that is both a regulator and a target of autophagy. Altogether, these resources provide a foundation for building new hypotheses regarding protein networks and function.

Protein-protein interactions (PPIs) are central to cell biological processes, such as formation of multiprotein complexes and enzymes, receptor-ligand and kinase-substrate interactions, intracellular signal transduction, and regulation of transcription and translation. A number of complementary methods can be used to identify PPIs, including mass spectrometry-based methods for identification of protein complexes and two-component methods such as yeast two-hybrid (Y2H) analysis for identification of binary interactions[1]. Results from systematically screened and validated binary interactions contribute to the development of specific hypotheses regarding the functional in vivo relevance of individual PPIs. Moreover, when applied at large scale and integrated with other datasets, networks of binary

A full list of affiliations appears at the end of the paper. ✉e-mail: celniker@fruitfly.org; marc_vidal@dfci.harvard.edu; perrimon@genetics.med.harvard.edu; stephanie_mohr@hms.harvard.edu

interactions elucidate new components of known pathways. Particularly relevant to this study, methods for identification of binary interactions have improved over the years and caveats to the approach are now well understood[2]. Innovations in experimental approach and analysis, as well as production of proteome-scale open reading frame (ORF) clone collections, made it possible to increase both the scale and the quality of binary interaction screens. Indeed, simply increasing the number of ORFs tested in Y2H assays contributes to new discoveries and brings protein-centric studies closer to the scale that can be accomplished with nucleic acid-based studies such as transcriptomics analyses. In recognition of the value of binary protein information to research study, binary interaction methods have been applied at an increasingly large scale for the discovery of PPI networks for several proteomes, including the human and yeast proteomes[3,4].

*Drosophila melanogaster* is an exemplary research system with a rich history of impactful contributions to our understanding of conserved biological processes and enduring relevance in biological and biomedical research[5–7]. The *Drosophila* research community has made significant investments in technology and resource development in addition to research studies, leading to a wealth of available genetic methods, fly stock reagents, large-scale datasets, and databases that can be used as research tools and mined for new hypothesis development, disease modeling, and experimental studies[8]. These include genome-wide genetic and RNAi screens[9–11]; extensive genomics studies[12,13]; large-scale transcriptomics studies for many *Drosophila* cell lines, developmental stages, and tissues[14–16]; large-scale studies of transcriptional regulation[17]; and single-cell RNAseq analysis[18,19].

Protein-based resources and datasets provide an important complement to other 'omics' resources but are hindered in scale by technological challenges. Nevertheless, several efforts have generated physical and data resources relevant to *Drosophila* proteins. The first attempt at generating a binary protein interactome map for *Drosophila* at proteome-scale was released two decades ago[20], followed by a few attempts at smaller scale[21–23]. In addition, the large-scale *Drosophila* Protein Interaction Map (DPiM) project, which used affinity purification followed by mass spectrometry (AP-MS), identified associations for ~5000 fly bait proteins[24], a project that was made possible by the systematic ORFeome cloning project of the Berkeley *Drosophila* Genome Project (BDGP)[25]. In addition, focused studies to detect specific Drosophila PPI networks have been reported, e.g., related to InR/PI3K/Akt signaling[26], Hippo signaling[27], Golgi phosphoprotein 3 (GOLPH3)[28], and GAGA factor[29]. Moreover, databases of known and predicted *Drosophila* PPIs have been established and updated, such as the *Drosophila* Interaction Database (DroID)[30–32] and databases with multi-species coverage, including the Molecular Interaction Search Tool (MIST)[33], BioGRID[34], and IntAct[35]. Nevertheless, discovery of high-confidence binary interactions using ORF collections and up-to-date methods has remained limited in *Drosophila*.

To address this unmet need, we applied to *Drosophila* the overall strategy for large-scale, high-confidence detection of binary protein interactions and data integration that was recently reported for the human proteome[3]. Our approach involved two distinct configurations of the Y2H assay for a total of four all-by-all Y2H screens of 10,000 × 10,000 *Drosophila* proteins and resulted in a new *Drosophila* binary interaction dataset, the "FlyBi" dataset, of 8723 binary interactions among 2939 proteins. Subsequent reanalysis of previous datasets and integration of FlyBi data and literature-based binary interactions of comparable quality resulted in an expanded, high-confidence *Drosophila* reference interaction (DroRI) network of 17,232 binary interactions among 6511 proteins. We tested the utility of the data to predict function by generating a putative autophagy interaction network that we validated in vivo using autophagy-related assays. The ORF clone collection and data resources generated in this project, available from multiple public sources, provide a foundation for additional proteomics studies and for the generation of new hypotheses regarding protein functions in *Drosophila*.

## Results

### Binary protein-protein interaction network for *Drosophila*

Performing Y2H screens with multiple, state-of-the-art versions of the assay can lead to increased high-quality coverage of binary interactions, as demonstrated by analyses that use existing knowledge as a benchmark for quality analysis (e.g., see ref. 3). We have demonstrated that both specificity and sensitivity of maps can be increased by improving any one of the four parameters of the 'empirical framework': i) completeness of the search space to be explored; ii) assay sensitivity; iii) sampling sensitivity; and iv) precision[36–39]. Since the *Drosophila* proteome contains ~13,900 confirmed or predicted protein-coding genes, the complete search space to be eventually explored is at least a 13,900 × 13,900 matrix of $1.9 \times 10^8$ combinations. The first systematic attempt was performed by screening ~10,000 baits against two cDNA libraries and a pool of ~10,000 ORF clones[20]. Limitations of the study include that only a single assay version was used; a limited number of replicate screens were performed; and gene annotations were of poorer quality than they are now. In that study, a subset of 4780 PPIs were reported as reaching acceptable quality levels[20].

To improve our knowledge of the *Drosophila* binary interactome network, we chose to perform four large-scale Y2H screens with a set of ~10,000 *Drosophila melanogaster* proteins of known sequence (see https://www.fruitfly.org/DGC/index.html and ref. 40): two screens in each of two different configurations differing in the position of the Gal4 activation domain (AD) fusion, i.e., N- or C-terminus, and in the overall level of exogenous expression, i.e., using either centromeric or two-micron based expression vectors[3]. These four "all-by-all" screens represented 400 million combinations of protein pairs (Fig. 1a, Supplementary Fig. 1). First pass pairs (FiPPs) identified in the primary Y2H screens were systematically retested in pairwise tests, followed by sequence confirmation. Altogether, we identified 1726 interaction pairs in assay version 1, screen 1; 1029 in assay version 1, screen 2; 3908 in assay version 3, screen 1; and 3509 in assay version 3, screen 2. In addition, to test of the quality of the experimentally identified putative binary protein pairs (see next section, "MAmmalian Protein-Protein …"), we established (i) a small, high-confidence positive reference set (PRS) based on the literature and filtered to include only pairs for which both ORFs are present in the FlyBi ORF collection (Supplementary Data 1), (ii) a random reference set (RRS) of the same size and with the same filter applied (Supplementary Data 1), (iii) a larger list of literature-curated binary pairs for which multiple lines of evidence support the interaction (at the time of the analysis, i.e., Lit-BM-16, Supplementary Data 2, or the most recent available, i.e., Lit-BM-20, Supplementary Data 3), and (iv) a list of literature-curated binary pairs for which only one line of evidence supports the interaction (Lit-BS, Supplementary Data 4), similar to what was done for the human reference interaction (HuRI) network[3,41].

The list of sequence-confirmed putative binary interactors resulting from experimental Y2H testing was supplemented by application of a computational approach to predict additional interactions based on the interactions identified in screens 1 and 2 (assay version 1). Different from other approaches previously applied to *Drosophila* (e.g., see ref. 42), our method, known as the L3 approach, is based on connectedness of proteins within a network (see Methods and ref. 43). Following application of the L3 approach to data from the screens 1 and 2, we experimentally tested the top 1000 computationally predicted pairs. Excluding 254 undetermined pairs (see Methods), 71%, 80 and 90% of the top 1000, 500 and 100 predictions were scored as positives, comparing to 13% for Lit-BM-16 pairs. Of the 533 positive

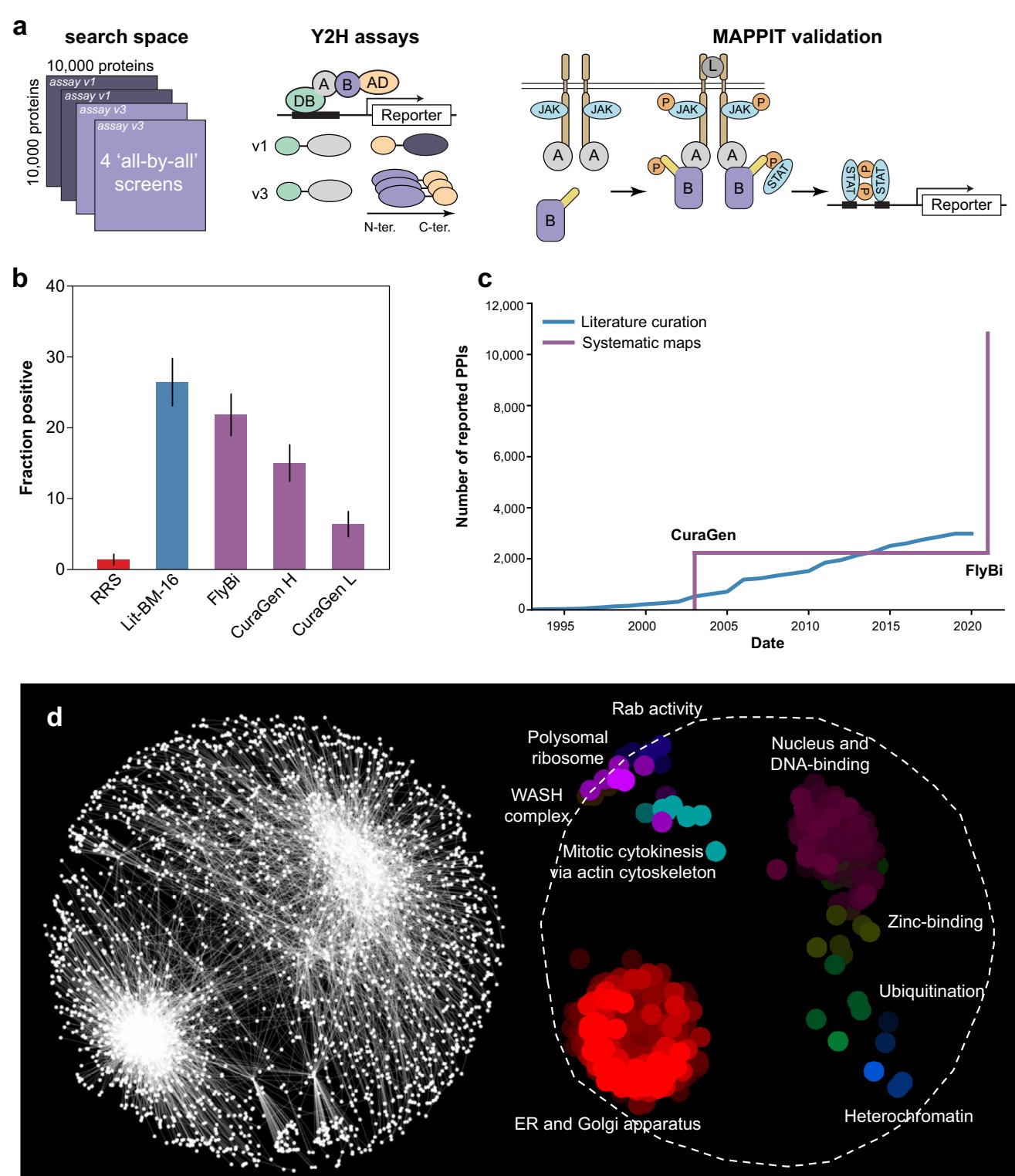

predictions, 332 interactions were sequence-confirmed and added to the FlyBi dataset.

Combining all pairs validated by pairwise testing results in a total FlyBi dataset of 8723 unique interaction pairs for 2939 genes. The FlyBi dataset is available as Supplementary Data 5 and the FlyBi project webpage (https://flybi.hms.harvard.edu/). In addition, these pairs have been integrated with other datasets at IntAct (https://www.ebi.ac.uk/intact/)[44] and in MIST (https://fgrtools.hms.harvard.edu/MIST/)[33].

**MAmmalian Protein-Protein Interaction Trap (MAPPIT) analysis**
Verifying putative binary interactions with orthogonal assays provides a method for quality analysis that can be used to define cut-off values prior to integration of data from different sources. Thus, our next goal was to analyze the quality of the FlyBi pairs and of Lit-BM pairs (available at the time of the experiments; Lit-BM-16; see Supplementary Data 2), as well as binary pairs from the literature with only a single piece of evidence (Lit-BS), pairs identified in the previous large-scale *Drosophila* Y2H study (CuraGen pairs)[20], binary pairs identified in

**Fig. 1 | Large-scale, all-by-all binary interaction screens of 10,000 *Drosophila* proteins. a** Schematic of the systematic screening pipeline. Left, the search space covered and the Y2H assay versions used. Center, assay versions used in the Y2H screen. Right, MAPPIT validation assay. **b** Fraction of pairs positive in the MAPPIT validation assay for the following sets: random reference set (RRS; *n* = 216; red), literature-curated binary pairs with multiple evidence (at the time of the assay; Lit-BM-16; *n* = 123; blue), FlyBi pairs (*n* = 193; purple), and CuraGen pairs at high (H; *n* = 193) and low (L; *n* = 187) cutoff values as defined by Giot, Bader et al. 2003 (purple). Error bars shown as fraction of positives ± standard error of the proportion. **c** Total number of binary interactions in literature and systematic interactome maps over the past 20 years. Blue line, gradual increase in the total number of binary interactions accumulated in the literature (Lit-BM), displayed based on date

of publication. Purple line, total number of interactions from systematic interactome mapping efforts based on the date of public release of systematic binary datasets. **d** Network-based spatial enrichment analysis (SAFE) results for the FlyBi dataset. Clusters of genes enriched for gene ontology (GO) terms are highlighted. The different colors highlight different enriched groups. Source data are provided in Supplementary Files 1–6. We note that panel **a** of this figure comprises modified versions of Fig. 1a, b in Luck, K., Kim, DK., Lambourne, L. et al. "A reference map of the human binary protein interactome." *Nature* 580, 402–408 (2020) https://doi.org/10.1038/s41586-020-2188-x and Fig. 2b in Braun, P., Tasan, M., Dreze, M. et al. "An experimentally derived confidence score for binary protein-protein interactions." *Nat Methods* 6, 91–97 (2009) https://doi.org/10.1038/nmeth.1281.

additional Y2H studies made available at DroID[31,32], and interactions identified in the DPiM project[24]. We experimentally tested randomly selected subsets of pairs from the FlyBi and other datasets using the MAmmalian Protein-Protein Interaction Trap (MAPPIT) assay[45]. With the MAPPIT assay, binary interactions between two proteins expressed in mammalian cells activate signaling by an otherwise inactive cytokine receptor. The lists of pairs tested in the MAPPIT assay and the test results are provided in Supplementary Data 6 (see also Methods and Supplementary Figs. 2, 3).

The results of this analysis made it possible for us to apply a cut-off value for CuraGen pairs that produced a list of pairs of equivalent high quality as compared with FlyBi pairs from assay version 1 (N-N terminal configuration) as judged by performance in this orthogonal MAPPIT assay. The Giot et al. study reports that 4,780 interactions among 4,679 proteins met the cut-off value of 0.5 for high-confidence as defined in that study[20]. We found that CuraGen pairs with a confidence score of 0.7 or higher as defined in ref. 20 have a similar recovery rate in the MAPPIT as compared with FlyBi pairs (Supplementar Figs. 2, 3). Thus, a total of 2,232 protein pairs from the CuraGen dataset met the quality cut-off criteria for integration into our final reference map as described below. We note that in the N-C terminal configuration literature pairs and pairs from the FlyBi assay 'version 3' screens validate at a lower level than literature pairs and 'version 1' screen pairs using the N-N terminal configuration (Supplementary Fig. 3). Literature pairs also did not validate at the same rate when tested using the C-terminal version of MAPPIT. This is also consistent with what we observed for testing of 'version 3' pairs from a human ORF screen[3]. We attribute this to the fact that the MAPPIT assay has not been optimized for screens performed using C-terminally-fused ORFs (N-C MAPPIT validation of N-C Y2H pairs performs better than N-N MAPPIT testing of N-C Y2H pairs but does not perform as well as N-N MAPPIT testing of N-N Y2H pairs). N-C MAPPIT validation of N-C Y2H pairs performing better than N-N MAPPIT testing of N-C Y2H pairs but not as well as N-N MAPPIT testing of N-N Y2H pairs also supports this conclusion. As expected due to significant differences in the assay formats, assay types, and other relevant factors, the positive rates with MAPPIT were lower for DPiM[46] and for the group of previous smaller-scale Y2H studies available from DroID[31,32] (Supplementary Fig. 3). These other studies contributed to defining the Lit-BM, e.g., as the source of additional evidence for some pairs, and notably, the Lit-BM performs significantly better than the Lit-BS (Supplementary Fig. 3). This provides one indicator among many that these other datasets have clear value as part of an effort to fully document PPIs in *Drosophila*.

**Comparison of FlyBi interactions with existing knowledge**

We next compared FlyBi pairs with interaction data from a variety of data repositories that are integrated in MIST[33] (Fig. 2). We generated 1000 randomized versions of the FlyBi network by node shuffling. Interacting pairs in the FlyBi dataset show significant overlap with physical interaction data obtained from previous studies in *Drosophila* and physical interactions mapped from orthologous genes

('interologs') (Fig. 2a). We also observed some overlap between FlyBi binary interaction pairs and genetic interaction (GI) data for *Drosophila* and between orthologs of fly genes in the budding yeast *Saccharomyces cerevisiae* (Fig. 2a). To further analyze FlyBi interactions, we determined the count of literature citations for each gene in the Lit-BM-20 or FlyBi dataset. As expected, interactors in Lit-BM-20 are biased towards well-studied genes (i.e., genes with larger numbers of literature citations). By contrast, we did not observe this bias for genes in the FlyBi dataset (Fig. 2b), consistent with the large-scale, all-by-all approach we took to generate the data. We next compared gene ontology (GO) annotations for the two proteins in each pair in three categories—biological process, molecular function, and cellular component—as well as phenotype annotations from FlyBase. For both Lit-BM-20 and FlyBi pairs, we observe significant enrichment for genes with the same GO and/or phenotype annotations as their interacting partners (Fig. 2c). We also compared binary interactions with protein complex-based interaction data, and with components of protein complexes as annotated in literature[47], and observed enrichment in both the Lit-BM-20 and FlyBi sets (Fig. 2c). In addition, interacting proteins identified in our study are more likely to be found in the same organelle and in the same cell type, as well as reported in the same publications, compared to the random controls (Fig. 2d–f).

Comparing the Lit-BM-20 and FlyBi sets to random networks reveals that FlyBi interaction pairs are of a quality that is comparable to the high-confidence published binary interactions that make up the Lit-BM-20 and are less biased. As such, these sets can appropriately be combined to generate a high-quality *Drosophila* reference interaction (DroRI) network. We built a new, high-confidence DroRI network by integrating the FlyBi data, CuraGen data that meet the cut-off for data quality equivalent to the FlyBi data, and all other high-confidence binary interactions (i.e., literature-based interactions). The DroRI network is comprised of 17,232 interactions among the protein products of 6511 genes and can be queried and accessed at a dedicated page at MIST. To facilitate integrated data mining and hypothesis generation, we integrated tissue-specific bulk RNA-seq data generated by the modENCODE consortium[48] into MIST. This makes it possible for users of MIST to project any of the tissue-specific transcriptomics datasets onto the DroRI network, and reveal the subset of network interactions predicted to occur in a given tissue (Supplementary Fig. 4).

We also compared the FlyBi network with interaction data from the mass spectrometry-based DPiM study, and with binary interaction maps from other species. We found that 72 pairs in FlyBi overlap with DPiM data, which represents 2% and 0.9% of interactions detected by Y2H and AP-MS methods, respectively, within the space of protein pairs tested in each study. We note that another study that used both AP-MS and Y2H, only 6 interactions (representing 1.3% of interactions detected by Y2H and 0.16% of interactions detected AP-MS) were detected by both methods[49]. We also note that the observed overlap between FlyBi and DPiM datasets is greater than expected if the assays were completely independent (Supplementary Table 1, Fisher's exact test, $P < 2.2e\text{-}16$, OR = 45.1). We further note that a total of 2661 pairs in

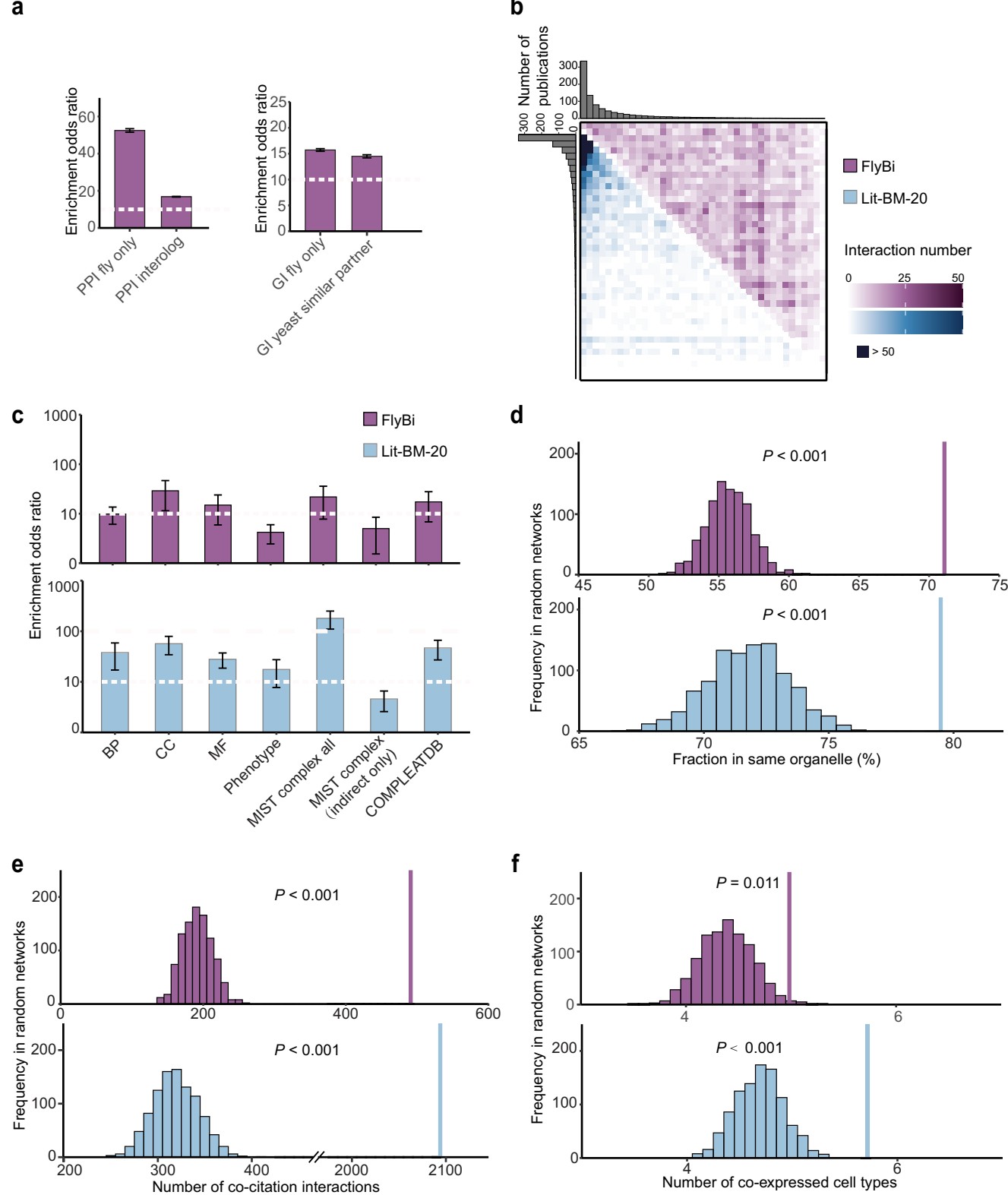

the DroRI overlap with the complete set of interactions detected by mass spectrometry as annotated in MIST.

With regards to other species, we find that comparison of the DroRI and HuRI networks reveals 714 of a total of 9332 interactions for which both orthologs are present in both datasets are identified as binary interactors in both networks. The total set of DroRI binary interactions for which orthologs are detected as binary interactors in any of the species included in MIST (human, rat, mouse, zebrafish, *X.*

*laevis, X. tropicalis, C. elegans, S. cerevisiae*, and *S. pombe*) is 1355. The low level of overlap likely also reflects differences in what is discoverable using different types of assays, whether or not the correct isoform is being tested, other sources of false negative discovery, and meaningful biological differences.

Ultimately, the value to the *Drosophila* research community of the DroRI network, and the new FlyBi dataset in particular, will be revealed by exploring its use, such as for the development of new hypotheses

**Fig. 2 | Bioinformatics analysis of the FlyBi Y2H dataset. a** Comparison of FlyBi with protein-protein and genetic interactions (PPIs and GIs) from MIST. Fraction of FlyBi pairs overlapping with published *Drosophila* PPIs or interologs (putative PPIs based on orthologous genes in other organisms) analyzed by comparison to 1000 random networks generated by node shuffling. FlyBi dataset is enriched for published PPIs. Overlap with *Drosophila* GIs and gene pairs with similar GIs in yeast were also analyzed. **b** Adjacency matrix for binary interactions in literature with multiple lines of evidence (Lit-BM; light blue) and FlyBi interactions (purple). Proteins binned and ordered along axes based on number of corresponding publications. Color intensity of each square reflects total number of interactions between proteins in corresponding bins. **c–f** Biological significance analyses. Purple, FlyBi; light blue, Lit-BM. **c** Enrichment of binary interactome maps for functional relationships and co-complex memberships. BP Biological process, CC Cellular component, MF Molecular function; Phenotype, shared phenotypes as annotated by FlyBase; MIST complex all, all annotated indirect interactions in MIST (might or might not be supported by direct evidence); MIST complex only indirect, all interactions

annotated as supported only by indirect evidence in MIST; COMPLEATDB, complex annotations (literature-based complexes only). Dashed white line, reference for comparison of bar heights. For **a, c**, means ± standard errors of the means (SEM) are shown. For **d–f**, single bar on right shows results for FlyBi (purple) or Lit-BM (blue); multiple bars on left show results for 1000 randomized networks. **d** Co-localization analysis. Shown, fraction of interacting partners sharing the same organelle annotation, compared with results for 1000 randomized networks. Organelle annotations as predicted by PSORT and DeepLoc. **e** Co-citation analysis. Shown are numbers of interacting partners cited in the same publication(s), compared with results for 1000 randomized networks. Only publications associated with <100 genes considered. **f** Co-expression analysis. Shown is average number of co-expressed cell types defined by cluster analysis of a single-cell RNAseq dataset for *Drosophila* midgut, compared to results with 1000 randomized networks. Statistical information is provided in Supplemental Notes. Source data are provided in Supplementary files 1–6.

regarding protein function. We describe the results of one such exploration below.

## Generation of an autophagy network using FlyBi data

To experimentally test the predictive value of interactions represented in the DroRI network and in particular, to test the predictive value of the new FlyBi binary interaction pairs with regards to shared gene function, we chose to focus on autophagy. Autophagy has been extensively studied in multiple species[50]; has been characterized using protein-centric approaches in human cells[51]; is a conserved process with human health relevance[52]; and is easily studied in vivo in *Drosophila* using multiple well-established assays[53,54]. To identify new regulators of autophagy, we used FlyBi data to define a list of candidate autophagy-related proteins and a control set. To build a putative autophagy-related list, we first assembled a list of 19 known autophagy regulators (List 1 in Supplementary Data 7). Next, we mined the FlyBi data for interactors with these autophagy regulators and identified 48 candidate interactors (List 2 in Supplementary Data 7). One of the 19 known regulators we included is Atg8a. There are five interactors with Atg8a in the FlyBi dataset. We note that two of these five were also identified in a recent Y2H screen for interactors with Atg8a[55]; i.e., Diabetes and obesity regulated (DOR), a known autophagy regulator, and CG12576, an uncharacterized protein. In addition, the DOR-Atg8a interaction was identified in the CuraGen Y2H study[20] and a physical interaction between the two was further confirmed by CoIP and characterized in another study[56]. To expand the candidate list, we again mined the FlyBi data, and identified 103 additional potential interactors of List 2 proteins (List 3 in Supplementary Data 7). By combining Lists 1, 2, and 3, we generated a putative PPI network related to autophagy that includes four core autophagy-related genes and 166 candidates (170 gene 'autophagy set') (Supplementary Data 8). As a control set, we chose at random 106 genes from the FlyBi dataset ('random set') (Supplementary Data 8).

To test for autophagy-related functions, we performed loss-of-function experiments using RNAi (Fig. 3a) combined with over-expression of *Atg1*, which encodes a protein kinase essential for autophagy (Fig. 3a). Overexpression of Atg1 in the Drosophila eye induces a high level of autophagy, leading to a rough eye phenotype (Fig. 3a, compare b' and A")[57,58]. To test the roles of the 'autophagy set' genes, we determined if RNAi knockdown of these candidates modified the *Atg1*-induced rough eye phenotype. A total of 477 RNAi lines targeting 166 genes were tested in a *GMR-Gal4 > UAS-Atg1* background. We found that 234 lines, corresponding to 137 genes, modified the severity of the *GMR-Gal4 > UAS-Atg1* phenotype (Supplementary Data 8). To address whether the data from FlyBi used to generate the autophagy network helped enrich for potential autophagy components, we randomly selected 106 genes from FlyBi as a control list of comparable size ('random set') and tested these genes in the *GMR-*

*Gal4 > UAS-Atg1* assay. Altogether, 26 of the lines tested (24%) modified the severity of *GMR-Gal4 > UAS-Atg1* phenotype (Supplementary Data 8). We tested multiple lines per gene in the autophagy set and only a single line per gene in the control set. Thus, to appropriately compare the percentage of modifiers between the autophagy set and the control set, we randomly selected one RNAi stock per gene from the autophagy set five times, generating five independent data subsets (see Methods) (Supplementary Data 8). RNAi lines tested in the autophagy sub-set modified the *GMR-Gal4 > UAS-Atg1* phenotype in 50%, 52%, 47%, 51%, and 55% of cases (average = 51%), compared to 24% in the random set (Fig. 3b). Altogether, these results indicate that the targeted candidate gene screen approach is more efficient at identifying new potential modifiers of autophagy-related functions. This is consistent with a previous report that showed that protein network information can be used to limit false discovery in *Drosophila* RNAi screens[59].

We next tested putative autophagy regulators identified in the *GMR-Atg1* screen using a different assay performed at a different life-cycle stage and in a different tissue (Fig. 3c). This assay interrogated autophagy-related processes in the larval fat body, a nutrient storage organ analogous to the human liver in which autophagy is quickly induced by starvation[60]. Under fed conditions, the autophagosomal marker mCherry-ATG8a shows is detected in a diffuse distribution throughout the cells (Fig. 3c). Upon starvation, mCherry-ATG8a redistributes to form punctate structures (autophagosomes) in the cytoplasm (Fig. 3c). Of the 234 RNAi lines identified in the *GMR-Gal4 > UAS-Atg1* eye screen, 60 (26%) increased fat body Atg8a puncta under fed conditions, while 41 lines (18%) inhibited fat body Atg8a puncta formation upon starvation (Supplementary Data 8). As an example of a negative autophagy regulator, depletion of *dwg* in GFP-labeled flip-out clones induced Atg8a punctate formation under nutrient rich conditions (Fig. 3c, top panels). In addition, as an example of a positive regulator of autophagy, depletion of *MED15* suppressed starvation-induced Atg8a puncta formation (Fig. 3c, bottom panels). Altogether, our candidate gene approach allowed us to quickly and efficiently identify new genes that are likely to be regulators of autophagy.

In total, 101 RNAi lines corresponding to 66 genes from the 'autophagy set' list were able to modify Atg1-induced eye defects and alter Atg8 puncta. Of the 66 genes, we chose 39 genes for which there are at least two RNAi lines available and results with both lines had consistent effects in both the fat body and the eye phenotype screens. We tested whether components of the autophagy network can physically interact in *Drosophila* cells by co-immunoprecipitation (Co-IP). We expressed Flag- and GFP-tagged proteins together in *Drosophila* S2R + cells, pulled down GFP-tagged proteins, and determined whether they were associated with Flag-tagged proteins (Supplementary Fig. 5, Supplementary Fig. 6, Supplementary Table 2). Of the 39 genes,

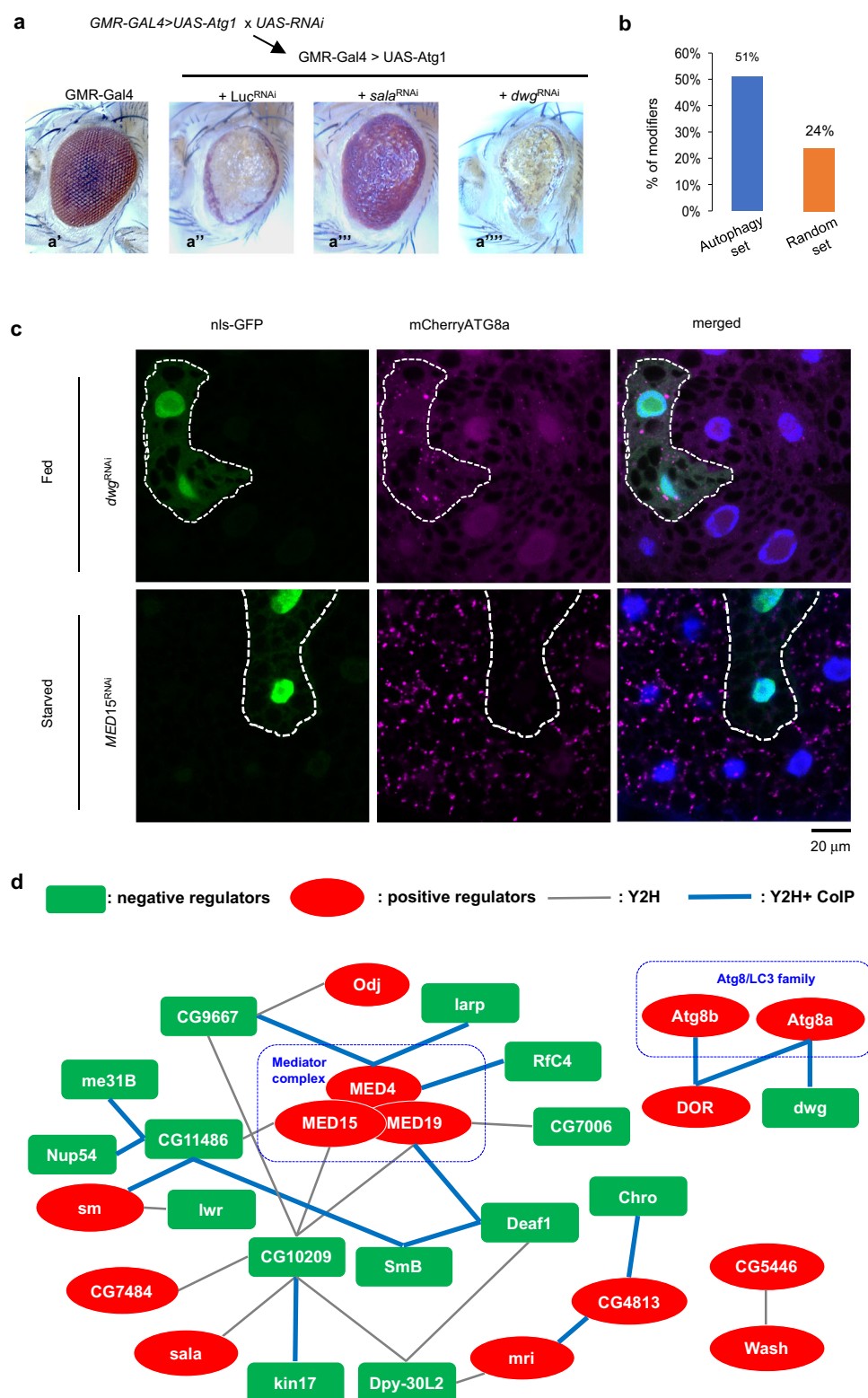

**Nature Communications** | (2023)14:2162

one of the GFP-tagged proteins, CG10209, was expressed at very low levels. To overcome this issue, we designed a smaller Flag-tagged form of CG10209 and enriched it using Flag-beads (Supplementary Fig. 7). Of 29 pairs we tested, an interaction was detected using co-IP for 16 (55%), providing support for the high quality of the FlyBi dataset. The observation that 55% of pairs were positive by co-IP is similar to what has been reported by others using co-IP as follow-up to Y2H studies in other species, e.g., successful co-IP of 5 of 12 (42%) in one recent report[61] and 45 of 79 (57%) in another[62].

## Dwg binds insulators near *ATG* genes to suppress autophagy

One of our candidates, *deformed wings (dwg;* also known as *Zw5)*, encodes a *Drosophila* insulator protein responsible for enhancer blocking and support of distant interactions, contributing to the organization of chromosome architecture[63]. Our genetic test showed that *dwg* is a putative negative regulator of autophagy. Consistent with this, whole larval lysate of *dwg* mutants showed a higher level of autophagy, indicated by increased lipidated Atg8a (Atg8a-II) compared to lysate from control (Fig. 4a).

**Fig. 3 | Identification of an autophagy regulatory network using the FlyBi dataset. a** Genetic cross and example phenotypes for RNAi knockdown in the presence of *Atg1* over-expression. Two sets were compared: an experimental set defined based on predicted interaction in the FlyBi dataset with known autophagy components or their interactions (Supplementary Data 8) and a randomly selected set. **a′-a′′′′.** Representative adult *Drosophila* eye phenotypes from control and experimental assays for modification of the *Atg1* overexpression phenotype. **a′,** Gal4-only control. **a′′,** Ectopic expression of *Atg1* using the eye-specific *GMR-GAL4* driver results in a rough eye and reduced eye size. The effect is reduced in the presence of *Sala^RNAi* (**a′′′**) and more severe in the presence of *dwg^RNAi* (**a′′′′**). **b** Percentage of RNAi lines that behaved as putative genetic modifiers of *Atg1* over-expression. **c** Distribution of mCherry-ATG8a in the larval fat body (fed or starved

conditions). Clonal expression of *dwg^RNAi* (GFP-labeled cells, top panels) induced formation of mCherry-ATG8a puncta under fed conditions whereas clonal expression of *MED15^RNAi* (GFP-labeled cells, bottom panels) abrogated starvation-induced Atg8a puncta. Experiments were repeated three times independently with similar results. Scale bar: 20 μm. **d** Putative autophagy regulator network based on knockdown, FlyBi data, and co-immunoprecipitation (co-IP) data (see Supplementary Figs. 5, 6). Green boxes, putative suppressors of autophagy; red ovals, putative inducers of autophagy; thin grey edges, direct interactions as reported in the FlyBi dataset; thicker blue edges, interactions reported in FlyBi and confirmed by co-IP. Of the genes in the network, 6 (30% of total) computed relatively unstudied genes (CGs) were added to the network by our studies. Source data are provided in Supplementary Data 8 and as a Source Data file.

We hypothesized that as an insulator, the Dwg protein might regulate autophagy through binding to insulator elements on chromatin and blocking enhancer functions. We therefore performed chromatin-immunoprecipitation followed by next-generation sequencing (ChIP-seq) to identify Dwg downstream targets. Gene group enrichment analysis revealed that the chromatin regions of autophagy-related genes and genes related to mitochondria, major signaling pathways, and ribosomes are targeted by Dwg (Fig. 4b). Interestingly, the Dwg-binding regions verified by ChIP-qPCR are located at or near insulator elements in four core ATG genes, Atg1, Atg3, Atg13, and Atg17 (Fig. 4c, d)[64]. Dwg can suppress enhancer functions, thus leading to inhibition of transcription[65]. Consistent with this, we also observed that *dwg* mutants showed higher mRNA expression of *ATG* genes (Supplementary Fig. 7). Taken together, these results suggest that Dwg binds to the insulator elements present in the *ATG* genes, presumably suppressing their transcription.

### Dwg is subjected to autophagy-lysosomal degradation

Autophagy is considered a highly selective pathway that targets specific substrates for degradation and selectivity is thought to rely mainly on the interaction between LC3/ATG8 family proteins and cargo/adaptor proteins[66]. Interestingly, our co-IP results suggest that Dwg physically interacts with Atg8a (Supplementary Fig. 8) and FlyBi data further suggest that the interaction is direct. These results suggest that Dwg is a substrate for autophagy. To test this hypothesis, we expressed Dwg in S2R + cells and treated cells with an autophagy inducer (Rapamycin) or a lysosomal inhibitor (Bafilomycin A1; BafA1). Immunoblots revealed that Dwg protein levels were reduced following Rapamycin treatment, whereas the Rapamycin-induced reduction of Dwg protein can be reversed by cotreatment with Bafilomycin A1 (Fig. 4e), indicating that Dwg is degraded by autophagy.

The mammalian ortholog of Atg8a, LC3, interacts with LIR (LC3-interacting region) motifs, W/F/Y-x-x-L/I/V, on substrates for autophagic degradation[66]. There are four potential LIR motifs predicted in Dwg (Supplementary Fig. 8)[67]. To characterize which LIR motifs are responsible for interactions with Atg8a, we generated four Dwg deletion mutants, Dwg-F1-F4. Each one of them contains an individual LIR motif (Supplementary Fig. 8). Our co-IP results showed that Atg8a interacts with Dwg-F1 and Dwg-F4 (Supplementary Fig. 8), suggesting that it is the first and fourth LIR motifs that bind to Atg8a. Consistent with this result, Dwg with mutant LIR motifs (Dwg^Y129A-I132A, Dwg^F401A-L404A, and Dwg^Y129A-I132A-F401A-L404A (4A)) had dramatically reduced interactions with Atg8a, demonstrating that these two LIR motifs are Atg8a binding sites (Fig. 4f).

### Atg8a delivers Dwg to autophagosomes for degradation

To elucidate the physiological role of the Dwg-Atg8 interaction, we expressed wild-type Dwg or Dwg with mutations in the two LIR motifs (Dwg^4A) and examined their localization and effects in S2R + cells and larval fat body. As expected, Dwg is localized in the nucleus (Fig. 4g and Supplementary Fig. 9). Inhibition of autophagosome degradation by Baf-A1 resulted in an increase of detectable Dwg in the cytoplasm and

co-localization of Dwg with Atg8a punctae in S2R + cells (Supplementary Fig. 9). Similarly, in the fat body, starvation induces translocation of Dwg to the cytoplasm, where it colocalizes with autophagosomes (Fig. 4g). These results further support that Dwg is a substrate of autophagy. Expression of Dwg with LIR motif mutations was restricted to the nucleus and strongly inhibited autophagy in both S2R + cells and fat bodies, suggesting that Atg8a is able to interact with and deliver Dwg to autophagosome for degradation (Fig. 4g and Supplementary Fig. 9). Altogether, our results suggest that disruption of the Dwg-Atg8a interaction not only stabilizes Dwg protein, but also allows Dwg to bind to insulator elements which suppress transcription of *Atg* genes, leading to autophagy inhibition (Supplementary Fig. 10).

## Discussion

In this work, we applied state-of-the-art experimental approaches to binary interaction mapping, together with experimental and bioinformatics-based quality analyses, to generate a next-generation reference binary interactome for *Drosophila*. The outcomes of our large-scale efforts include (i) a collection of ~10,000 *Drosophila* ORFs in a Gateway-system entry vector; (ii) a new high-confidence Y2H dataset, the FlyBi dataset, which is comprised of 8723 binary interactions among 2939 proteins; and (iii) an integrated *Drosophila* reference interactome, DroRI, which is comprised of 17,232 interactions among 6511 proteins. Features that distinguish the FlyBi project from past efforts include the quality and coverage of the ORF collection on which we based our Y2H screens;[25] use of improved versions of the Y2H system;[3] computational prediction of new interactors using a different approach than had previously been applied in *Drosophila*;[43] use of an orthogonal experimental approach to define cut-off values for confidence for FlyBi Y2H data, computational predictions, and previously reported data;[45] and integration and comparison of FlyBi data with existing PPI and other datasets to generate the DroRI resource that can be navigated using MIST[33].

Several indicators point to the value and high quality of FlyBi interaction pairs. For example, proteins in FlyBi pairs are less biased towards well-understood proteins as judged by the number of publications per gene as compared to existing pairs (Fig. 2b), such that they provide an important supplement to existing interaction datasets. Moreover, FlyBi pairs also performed as expected for a dataset enriched in biologically meaningful associations (Fig. 2c–f). Nevertheless, the pairs defined in this work have little overlap with interactors as defined using mass spectrometry-based approaches (e.g., the DPiM dataset) or with binary interactors observed in other species, observations that likely reflect both experimental and biological differences. Ultimately, the value of identification of large-scale binary datasets for biological and biomedical research lies in the ability to use individual identified putative PPIs and/or integrated networks to build new hypotheses that lead to efficient detection of new functional findings with in vivo relevance. To test this, we explored potential new interactors of proteins previously identified as relevant to autophagy in *Drosophila*. We chose to focus on autophagy because this process is well-studied in multiple species and has human health relevance, and

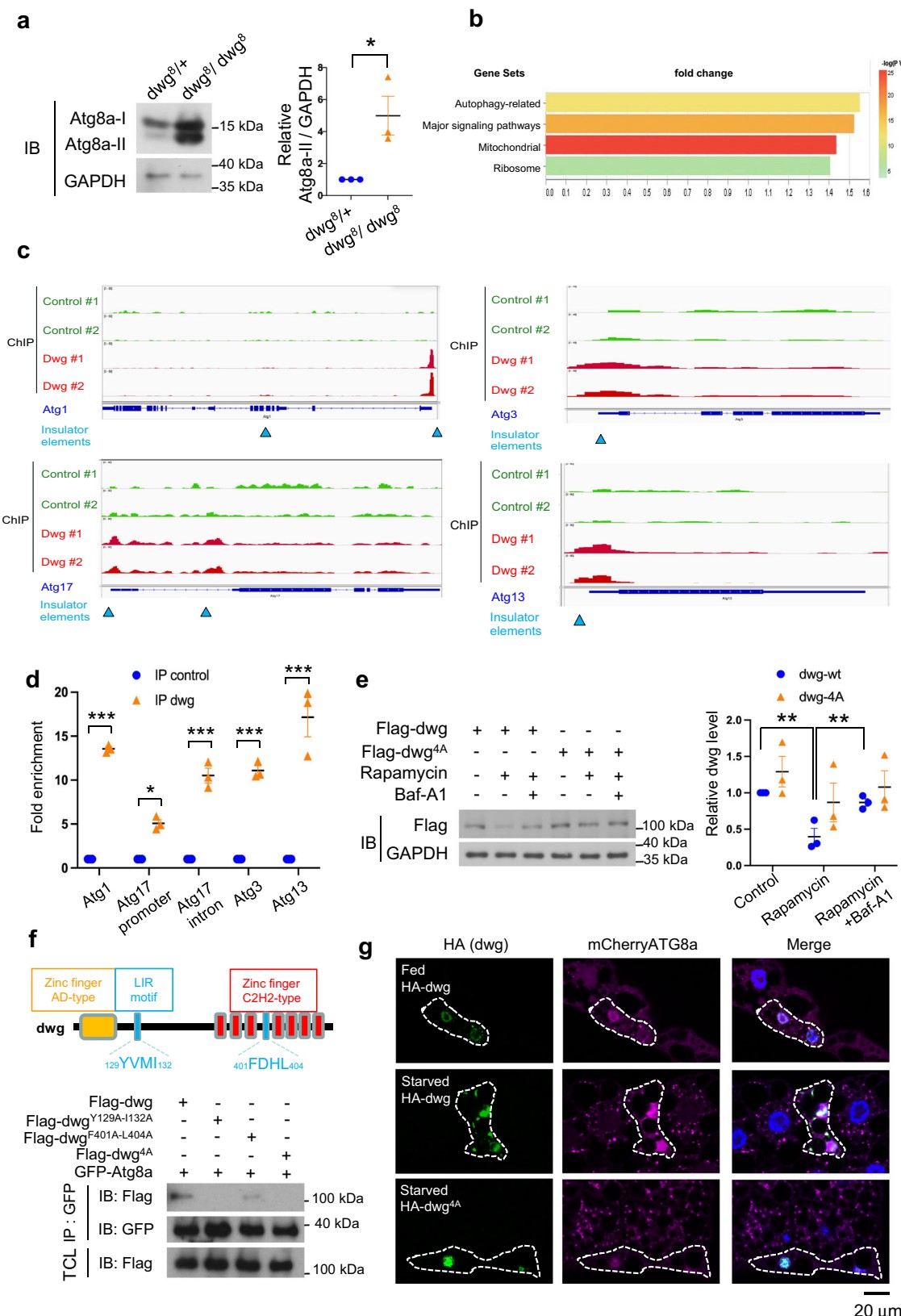

because well-established in vivo *Drosophila* assays related to autophagy were available.

Our approach was to start with known proteins of the autophagy pathway, identify potential PPIs based on the FlyBi data, and test these candidates for autophagy-related phenotypes in *Drosophila* using two different in vivo assays. We performed a focused RNAi screen for

putative genetic modifiers of the mild phenotype associated with overexpression in the eye of *Autophagy-related 1 (Atg1)*[57,58]. Using the positive hits from the *Atg1* modifier screen we determined the distribution of fluorescent protein-tagged Atg8 in the *Drosophila* fat body under fed and starved conditions[68]. Following this approach, we identified a high-confidence sub-network of putative autophagy

**Fig. 4 | Dwg is both a negative regulator and a substrate of autophagy. a** Flies homozygous mutant for a loss-of-function allele of *dwg* exhibit increased autophagy. Control (*dwg*[8]/+) and mutant (*dwg*[8]/*dwg*[8]) larva subjected to immunoblotting with indicated antibodies. Measurements are mean ± SEM of triplicate experiments. Significance determined by two-tailed t-test. *p = 0.031. **b** Gene group enrichment analysis of ChIPseq data. Bar length, fold change in enrichment. Colors, strength of significance (*p*-value of -log10 for each term). **c, d** Dwg binds insulator regions of *Atg* genes. **c** Example browser images for *Atg1, Atg3, Atg13,* and *Atg17* from ChIP-seq in S2R + cells expressing Flag-Dwg. Aggregate data from two independent experiments shown. Blue triangles, insulator binding regions. **d** Dwg occupancy at or near insulator regions of *Atg* genes revealed by ChIP-qPCR. One-Way ANOVA followed by Tukey's multiple comparison test to identify significant differences; shown are means ± SEM of three independent experiments; ***P < 0.0001, *P = 0.039. **e** Autophagic activity regulates Dwg levels. S2R + cells transfected with *Flag-dwg or -dwg*[AA(Y129A-I132A-F401A-L404)]) were treated with Rapamycin (autophagy inducer) or Bafilomycin A1 (Baf-A1; lysosomal inhibitor). Dwg and GAPDH levels analyzed by immunoblotting (IB) with indicated antibodies and quantified. One-Way ANOVA

followed by Tukey's multiple comparison test to identify significant differences; shown are means ± SEM of three independent experiments; **p < 0.01; p = 0.0026 (*dwg*-wt control v.s. Rapamycin). p = 0.0089 (*dwg*-wt Rapamycin v.s. Baf-A1). **f** Mapping Dwg-Atg8a interaction sites. Schematic of domain structures and LIR (LC3-interacting region) motifs of Dwg. S2R + cells transfected with *GFP-Atg8a, Flag-tagged dwg,* or *Flag-tagged dwg* with mutations in LIR motif (*dwg*[Y129A-I132A], *dwg*[F401A-L404A], or *dwg*[4A(Y129A-I132A-F401A-L404)]) for 48 h followed by immunoprecipitation with anti-GFP nanobody. Immunoprecipitated proteins and total cell lysates analyzed by immunoblotting with indicated antibodies. **g** Disruption of Dwg-At**g**8a interaction inhibits autophagy. Clonally expressed HA-tagged Dwg is nuclear under fed conditions (green); upon starvation, it is detected in cytoplasm and co-localizes with autophagosomes labeled by mCherry-Atg8a (magenta). A version of Dwg with mutations in Atg8a-binding sites (*dwg*[4A]) remains nuclear and inhibits autophagy. Fat body cells stained with DAPI (blue). Experiments repeated three times independently with similar results. Scale bar: 20 μm. Source data are provided as a Source Data file.

---

regulators (Fig. 3) and found that Dwg both regulates and is regulated by autophagy, providing evidence of reciprocal regulation between autophagy and chromatin regulators. Moreover, as expected for high-quality data, these findings show that using the interaction network constructed with FlyBi data allowed us to enrich for genes relevant to the process of autophagy. The ability to use binary interaction data to reduce the full set of *Drosophila* genes to a subset of high-confidence candidates prior to in vivo phenotypic analyses, which can be both time- and resource-intensive, will unquestionably accelerate future studies.

## Methods

### Generation of a large-scale ORF clone resource
The entry clone collection was generated from 11,687 BDGP cDNA gold clones[40] (see https://www.fruitfly.org/EST/gold_collection.shtml) using attB-tailed PCR. See below for detailed descriptions of primer design and PCR amplification. The PCR products were quality controlled by detection and sizing on agarose gels as follows. PCR products were loaded into 1% (w/v) agarose gels (3 g Agarose, 300 mL 1xTAE) and run in 1xTAE buffer with New England Biolabs 1 kb ladder. Gels were imaged using a BioRad GelDoc XR system. Band sizes were calculated using BioRad Quantity One software (version 4.6.9). High-quality PCR products were cloned into the pDONR223 expression and cloning vector using BP Clonase. See below for a detailed description of the cloning protocol. Clones were stored as glycerol stocks prior to their use to generate yeast expression clones.

**Primer Design.** PCR primers were designed using Primer3 release 0.9[69] and the sequence of the open reading frames. The parameters applied were as follows: PRIMER_OPT_SIZE, 17; PRIMER_MIN_SIZE, 15; PRIMER_MAX_SIZE, 20; PRIMER_OPT_TM, 60; PRIMER_MIN_TM, 40; PRIMER_MAX_TM, 95; PRIMER_OPT_GC_PERCENT, 50; PRIMER_MIN_GC, 0; PRIMER_MAX_GC, 100; PRIMER_EXPLAIN_FLAG, 1; PRIMER_MAX_POLY_X, 18; PRIMER_SELF_ANY, 30; PRIMER_SELF_END, 30. For dicistronic cDNAs, the CDS to which primers were designed was chosen manually with preference given to the longer CDS. Primer sequences are provided in Supplementary Data 9.

**PCR amplification of ORFs.** Templates were inoculated from the Berkeley *Drosophila* Genome Project (BDGP) Gold Collection into 1.2 mL 2x YT medium with appropriate antibiotic (Chloramphenicol at 100 μg/ml final conc., or Carbenicillin at a final concentration of 100 μg/ml). Cultures were grown overnight (16–18 h) at 37 °C at 300 rpm. The overnight culture was diluted 1:10 with sterile water. PCR primers were purchased desalted and resuspended in Tris-EDTA (TE) buffer at 20 μM concentration from Invitrogen. Pairs of primers were combined and diluted with Milli-Q water to a concentration of 1.25 μM

(each primer). PCR reactions were performed using 5 μL Phusion HF Buffer (5X concentration), 0.5 μL dNTP (10 mM each, New England Biolabs), 5 μL primer pair mix (1.25 μM each primer), 3 μL template (1:10 cell dilution), 0.25 μL Phusion DNA Polymerase (New England Biolabs), and 11.25 μL sterile Milli-Q water, for a total reaction volume of 25 μL. Touchdown PCR[70] was performed with the following cycling parameters: 98 °C for 1 min; 5 cycles of (98 °C for 10 s; 56–46 °C, decrease by 2 °C each cycle; 72 °C for 7.5 min); 15 cycles of (98 °C for 10 s; 72 °C for 7.5 min); 72 °C for 10 min; 4 °C hold.

**BP Clonase reactions.** BP Clonase reactions were performed in a total volume of 5 μL, consisting of 1 μL 5X BP Reaction Buffer, 1 μl pDONR223 vector (75 ng/μL, uncut), 1 μL BP Clonase, and 2 μL of attB-tailed PCR product. BP reactions were incubated at 25 °C for 18 h. Immediately following incubation, 2 μL of the BP reaction was transformed into 10 μL of chemically competent *E. coli* DH5-alpha cells (prepared in-house). The mixture was incubated for 30 min on wet ice, heat shocked for 40 s at 42 °C, and incubated for 2 min on wet ice. Finally, 90 μl of SOC medium was added and the transformations were incubated for 1 h, 225 RPM, 37 °C in an orbital shaking incubator. The entire transformation reaction was inoculated into 1 mL LB/spectinomycin (100 μg/mL) and incubated 16–18 h at 37 °C, 300 rpm. Reactions and transformations were performed in 96-well format in standard thermal cycler plates. Glycerol frozen stocks (15% glycerol) were made by mixing 50 μL glycerol (30%) with 50 μL overnight culture.

### Amplification of ORFs for transfer to expression vectors and sequence analysis
We used PCR to amplify the ORFs from the large-scale ORF collection to generate a product that was used for cloning into the yeast expression vectors (see below) and useful for sequence analysis. PCR was performed using individually indexed 96-well M13 forward primers (Life Technologies) and a non-indexed M13 reverse primer (5'-GTAACATCAGAGATTTTGAGACAC-3'). The same amount of each amplicon from each plate was pooled as a single sample. Samples from each entry plate were sequenced using the Illumina platform. Sequencing reads were deconvoluted to the individual well level based on a combination of the 96-well index and the Illumina sample index, and by alignment to ORF sequences. A clone was deemed 'sequence confirmed' if a majority of the reads from the well (>10 reads) aligned to the expected ORF sequence. Only entry clones that were sequence confirmed were re-arrayed and used for further processing.

### Preparation of Y2H expression clones from the large-scale ORF clone resource
Using the M13 PCR product from the entry ORFs, we performed a LR reaction into pDEST-DB, pDEST-AD-*CYH2* (assay version 1) and pDEST-

AD-AR68 (assay version 3) using Gateway Technology (Invitrogen). Attributes of these plasmids are summarized in Supplementary Table 3. The DNA was isolated using a liquid handling robot (Qiagen 96-well Miniprep). DB ORF fusions were transformed into yeast strain Y8930 *(MATα), trp1-901 leu2-3,112 ura3-52 his3-200 ade2-101 gal4Δ gal80Δ cyh2r GAL2::ADE2 GAL1::HIS3@LYS2 GAL7::LacZ@met2*, and AD ORF fusions into yeast strain Y8800 *(MATα), trp1-901 leu2-3,112 ura3-52 his3-200 ade2-101 gal4Δ gal80Δ cyh2r GAL2::ADE2 GAL1::HIS3@LYS2 GAL7::LacZ@met2*.

### Y2H auto-activator identification and removal
Prior to the screen, haploid DB ORFs were spotted on SC-Leu-His media to test for auto-activation of the GAL1::HIS3 reporter gene in the absence of an AD-ORF plasmid. DB ORFs that grew on SC-Leu-His were removed from the collection.

### Y2H Screening
Large-scale Y2H screens were performed using two assay formats[3]. For the first two screens (assay version 1), pools of 1000 ORFs as preys in pDEST-AD-*CYH2* were screened against single pDEST-DB ORF baits. Both AD and DB are fused to the N-terminus of the ORF and expressed from yeast centromeric plasmids (Fig. 1a, center panel, "v1"). For screens 3 and 4 (assay version 3), we used preys in pDEST-AD-AR68, in which the AD is fused to the C-terminus of the ORF and plasmid copy number reflects use of a 2-micron origin instead of the yeast centromeric chromosome. We used assay version 1 prey constructs and tested these against same baits (Fig. 1a, center panel, "v3"). A detailed workflow is provided in Supplementary Fig. 1b and Supplementary Methods, and follows what was reported for[3]. Briefly, following inoculation of DB and AD ORFs in selective media and overnight culture, 10 ul of each DB was mated against 5 ul of a pool of 1000 AD's (kilopool). After an overnight incubation at 30 °C in liquid rich medium (YEPD), 10 ul of the culture was transferred into synthetic complete media (SC) without leucine or tryptophan (SC-Leu-Trp) to select for diploids. The following day, the culture was spotted on SC-Leu-Trp-His +3AT solid media to select for diploids in which the *GAL1::HIS3* reporter gene was activated. In parallel, diploid yeast cells were transferred onto SC-Leu-His+3AT solid media supplemented with 1 mg/l cycloheximide (CHX) for assay version 1 or 10 mg/l CHX for assay version 3. After 72 h incubation at 30 °C and one additional day at room temperature, we picked colonies that grew well on 3AT plates and did not grow on CHX plates.

### Yeast colony sequencing
To identify both bait and prey proteins for thousands of positive colonies, we used a method called SWIM-seq (Shared-Well Interaction Mapping by sequencing) as described[3]. Briefly, DB and AD-ORFs were simultaneously amplified from 3 μl yeast lysate, using well-specific primers. After PCR amplification, barcoded PCR products from an entire 96 well plate were pooled together and purified and sequenced with Illumina Solexa technology allowing for identification of interacting first pass pairs of proteins (FiPPs). To identify likely true AD/DB pairs, we developed a "SWIM score"[3] *S* that takes into account the AD and DB reads in each well, total reads returned from the sequencing run, and other factors. This is shown in Formula 1:

$$S = \frac{2}{\frac{a+M}{x} + \frac{d+N}{y}} \tag{1}$$

where *x* and *y* are read counts of an AD-ORF and DB-ORF in a given well respectively, *a* and *d* are total read counts of all aligned AD-ORF and DB-ORF in that well, and *M* and *N* are pseudo-counts for AD and DB respectively, which were constant for each sequencing batch but varied for different batches. We selected FiPPs for pairwise testing using a cutoff that balances the risk of testing too many false positives

FiPPs versus not testing too many true positive FiPPs. The cutoff varied for different screens and sequencing runs to adjust for slight variations in the screening and sequencing protocol. Primers used for SWIM-seq are shown in Supplementary Table 4.

### Pairwise test of FiPPs
Each FiPP was subjected to a pairwise retest[3]. Briefly, Ads and DBs were picked from the yeast ORF expression collection, mated in individual quadruplicates, and diploid yeast were spotted on selection media: SC-Leu-Trp-His+3AT and SC-Leu-His+3AT + CHX. Positive pairs were picked into SC-Leu-Trp and a SWIM-PCR was performed on the diploid yeast lysates to confirm the identity of ORFs. We used the SWIM score as described in ref. 3 to generate a list of binary interactors identified in the screens. If a DB acted as a de novo auto-activator, it was retested in a final pairwise experiment, where in parallel to mating the protein pairs, each DB was also mated against an "AD-null" plasmid without any ORF in the cloning site. Genes corresponding to mated yeast that grew on selective media when mated against AD-null yeast were removed from the final FlyBi dataset. A protein pair was scored as positive only when significantly more growth was observed on the test plate compared to the CHX plate. In the case of too strong growth on CHX plate, a pair was scored as auto-activator (classified as undetermined). If there was no growth on test and CHX plate, the pair was scored as negative. A pair was scored undetermined (NA) if the well was unscorable (contaminated, not spotted, etc.).

### Computation-based prediction of positive pairs and quality analysis
Computational prediction of positive pairs, based on the assay version 1 results (i.e., screens 1 and 2) was performed as described in[43]. We used network-based link prediction to rank candidate interacting pairs based on the normalized number of length three network paths linking them (L3). As the input, we used a list of 2195 PPIs from screens 1 and 2 and obtained the top 10,000 predictions. To quality-analyze these predictions and screen 2 data, we experimentally tested the top 1000 predictions from the L3 computational analysis, a set of 135 positive interactions from screen 2 (positive benchmarks), 263 proteins from the RRS (negative benchmarks), and binary interaction pairs from the following sources or lists: Lit-BM-16 (see main text) and DPiM[46]. Altogether, we pairwise tested 3399 non-redundant pairs in two orientations, allowing us to classify each pair as either positive, negative, or undetermined, following the experimental protocol described above and in Supplementary Methods.

### MAPPIT validation
MAPPIT analysis was performed as previously reported[71]. Entry clones for bait and prey proteins were first cloned into MAPPIT vectors via Gateway LR reaction. Miniprep DNA was used to transfect HEK293T cells by standard calcium precipitation in quadruplicate. For each tested pair, two wells were left untreated and two were stimulated with the cytokine erythropoietin (Epo), which can induce JAK-STAT pathway signaling in cells in which there is a bait-prey interaction, resulting in activation of a STAT-responsive firefly luciferase reporter[71]. MAPPIT validation assays were only deemed valid if both bait and prey were successfully cloned into expression vectors and bait expression was detected. Fold-induction values (i.e., the signal from stimulated cells / signal from unstimulated cells) were calculated for each pair and two negative controls (i.e., no bait with prey and bait with no prey). Each tested pair was assigned a quantitative score comprising the fold-induction value of the pair divided by the maximum fold-induction value of the two negative controls. The validation was done in several batches, and the same ~150 pairs of Lit-BM-16 (positive controls) and ~200 RRS pairs (random controls) were included in each batch. Altogether, we validated 844 screen pairs, as well as 193 CuraGen high confidence pairs, 187 CuraGen low confidence pairs[20], 216 pairs

reported in DPiM[46], 291 pairs in the "Finley Yeast Two-Hybrid Data" list that can be downloaded from the DroID online resource[31], and 187 Lit-BS pairs.

Pairs were scored positive or negative based on thresholds set at the 99th percentile of the RRS scores (equivalent to a 1% false discovery rate). Each experimental batch was scored separately and used the quantile function in the Python library. Pairs without valid quantitative scores were dropped, and recovery rates were calculated as the number of positive pairs over the sum of the positive and negative pairs. The error bars on the recovery rates were standard errors of the portions.

## Bioinformatic analyses

**SAFE analysis.** For SAFE analysis, we used the SAFE software[72] v1.5 to determine and visualize significant functional modules in various networks. The network layouts were generated with Cytoscape[73] v3.4.0 using the edge-weighted spring embedded layout. SAFE analysis was run using the default options with the exception that "layoutAlgorithm" was set to "none" (using the layout as generated by Cytoscape) and the "neighborhoodRadius" was set to "2."

**Comparison of FlyBi and DPiM.** To establish the screening space for DPiM, baits were extracted from a list provided by the authors[24] and prey proteins were defined as all expressed genes (FPKM > 0) in S2R + cells based on previous RNA-seq measurements[74]. To reflect the changes in gene annotations since the release of datasets, FBgn IDs were updated and validated using <http://flybase.org/convert/id>[75]. Genes corresponding to multiple updated and validated IDs were excluded. For the Fisher exact test, protein pairs present in the DPiM set but absent in the bait and/or prey lists (e.g., due to sensitivity of RNAseq) were added to the total search space for DPiM (-0.01% increase in search space size). Both FlyBi and DPiM screening spaces were filtered to include only annotated protein-coding genes, which were further filtered to unique pairs in an orientation-independent manner (i.e., Bait A – Prey B and Bait B – Prey A were counted as a single protein pair).

**Gene set enrichment.** Gene set enrichment analysis of genes covered by the FlyBi network was done using an in-house program written based on a hypergeometric distribution test. Gene sets were built based on the Gene List Annotation for *Drosophila* (GLAD) database[76]. A negative control of 1000 random networks was generated by shuffling FlyBi gene nodes 1000 times (node degree not necessarily preserved).

**Identification of Lit-BM-20.** Lit-BM-20 was built by selecting *Drosophila* physical interactions from MIST for which either the interaction was identified using one detection method for direct physical interaction as reported in multiple publications or the interaction was identified using multiple methods for detection of direct interactions (or both). The list of the detection methods for direct interaction were annotated based on the same criteria used for building the HuRI network[3,41]. Annotations of detection methods for interactions included in MIST were based on the European Bioinformatics Institute (EBI) molecular interaction (MI) controlled vocabulary system ([https://www.ebi.ac.uk/ols/ontologies/mi]), for example, MI:0800 for two hybrid.

**Comparisons to interologs.** The FlyBi dataset was compared with PPIs and genetic interactions detected in *Drosophila* and interologs as assembled by MIST. In addition, the FlyBi dataset was compared with *Drosophila* orthologous gene pairs mapped using DIOPT[77] from yeast gene pairs with similar genetic interactors[78].

**Adjacency matrix.** An adjacency matrix for binary interactions was built using FlyBi interactions and Lit-BM-20 interactions to visualize how frequently interacting proteins are reported in literature. The interacting proteins were binned and ordered along both axes based on the number of corresponding publications. The color intensity of each square reflects the total number of interactions between proteins in the corresponding bins.

**Gene Ontology (GO) analysis.** Analysis of the biological relevance for interacting proteins was done by evaluating commonality in the GO annotation, phenotype annotation, and complex memberships of the interacting proteins from FlyBi and Lit-BM-20 data as compared with protein pairs from the random networks. To do this, gene2go and gene2phenotype annotations were obtained from FlyBase[79], and GO terms with more than 30 associated genes were removed prior to enrichment analysis.

**Protein complex-based analysis.** Complex-based interaction data for *Drosophila* were obtained from MIST[33]. Protein complex annotations were obtained from COMPLEAT[47]. COMPLEAT includes annotated complexes from the literature and complexes predicted based on the connectivity of protein-protein network; however, only literature-based complexes were used for enrichment analysis.

**Co-localization analysis.** Co-localization analysis was done based on organelle prediction by PSORT[80,81] and DeepLoc[82]. Co-citation analysis was done based on associated literature for each interacting protein. Genome-scale studies were removed and only publications with fewer than 100 associated genes were considered. Co-expression analysis was done by mining a single-cell RNA-seq dataset for the *Drosophila* midgut[83] to identify cell types in which each interacting protein is expressed. The results were visualized by plotting the fraction of interacting pairs that share the same organelle annotation, the number of interacting pairs cited in the same publication(s), or the average number of co-expressed cell types of the interacting pairs, in each case as compared to results with the 1,000 randomized networks.

## Drosophila strains

Flies were raised at 25 °C following standard procedures unless otherwise noted. The following *Drosophila* strains were used: GMR-GAL4 (II) driver line (Perrimon lab collection), hs-flp; r4-mCherry-Atg8a Act > CD2 > GAL4 UAS-GFP-nls (mCherry-Atg8a; gift from Thomas Neufeld; described in[68]), w[1118] (Bloomington Drosophila Stock Center ID BDSC3605), y[1] w[*]; P{w[+mC]=UAS-Atg1.S}6B (UAS-Atg1; BDSC51655), dwg[8]/FM7a/Dp(1:3:Y)w[+] (dwg[8]; BDSC4094), y[1] v[1]; P{y[+t7.7] v[+t1.8] =TRiP.JF0135S}attP2 (UAS-Luc-RNAi; BDSC31603). The following strains were newly generated for this study: w[1118]; p{w + UAS-dwg-HA} and w[1118]; p{w + UAS-dwg-4A-HA}. The background genotypes of RNAi fly stocks used for the RNAi screen are as follows: y[1] v[1]; P{y[+t7.7] v[+t1.8] TRiP-shRNA}attP2 or attP40 (Transgenic RNAi Project [TRiP] lines); P{KK-RNAi}VIE-260B/CyO or /TM3 (Vienna Drosophila Research Center [VDRC] 'KK' lines); w[1118]; P{GD-RNAi}/CyO or /TM3 (VDRC 'GD' lines); and UAS-RNAi/CyO or /TM3 (National Institute of Genetics Japan [NIG-Japan] lines). Specific RNAi *Drosophila* stocks used for the genetic screen are listed in Supplementary Data 8.

## Antibodies

Antibodies were used for immunofluorescence (IF) or immunoblotting (WB) at the following dilutions. Rabbit polyclonal anti-GFP (A-6455, Molecular Probes), dilution factor 1:5000 (WB); rabbit monoclonal anti-Atg8 (ab109364, Abcam), dilution factor 1:2000 (WB) or 1:100 (IF); mouse monoclonal anti-Flag (F3165, Sigma), dilution factor 1:5000 (WB) or 1:1000 (IF); mouse monoclonal anti-HA (901514, Biolegend), dilution factor 1:1000 (IF); rabbit polyclonal anti-GAPDH (GTX100118, GeneTax), dilution factor 1:10,000 (WB); goat anti-Mouse IgG (H + L) secondary antibody, Alexa Fluor 633 (A-21052, Invitrogen), dilution factor 1:1,000 (IF); donkey anti-Mouse IgG (H + L) Secondary Antibody,

Alexa Fluor 555 (A-31570, Invitrogen), dilution factor 1:1,000 (IF); rabbit Anti-Mouse IgG (Light Chain Specific) (D3V2A) mAb (HRP Conjugate) (58802, Cell Signaling), dilution factor 1:1000 (WB); mouse Anti-Rabbit IgG (Light-Chain Specific) (D4W3E) mAb (HRP Conjugate) (93702, Cell Signaling), dilution factor 1:1000 (WB). For immunoprecipitation, we used a GFP Nanobody/VHH coupled to agarose beads (ChromoTek GFP-Trap Agarose, AB_2631357). For ChIP-seq, we used anti-Flag (Sigma, F3165) and the IgG antibody beads as included in SimpleChIP Plus Enzymatic Chromatin IP Kit (Cell Signaling Technology, 9005).

### Autophagy-related assays

**In vivo autophagy assay in adults and comparison of autophagy and random sets.** To compare datasets of comparable size, 106 nodes from a randomly generated network were selected as a control gene set and one stock per gene was screened for the modifier of *Atg1* overexpression-induced eye phenotypes. Specifically, we crossed TRiP, VDRC, or NIG-Japan RNAi fly stocks to *GMR-Gal4 UAS-Atg1* flies, then scored the eyes of adult males no older than two weeks post eclosion for eye phenotypes (we looked at males only to avoid differences in eye size due to sex). To compare the autophagy set covered by multiple stocks per gene with the random set where only one stock per gene was used, we randomly selected one stock per gene for the autophagy set and compared it with the result from the random set. We repeated this comparison process five independent times.

**In vivo autophagy assay in larvae.** Second instar larvae were collected 72–96 h after egg laying and cultured in fresh fly media with yeast paste (fed) or in vials containing 20% sucrose (starved) for 4 h. Autophagy level is indicated by autophagosome numbers labeled by mCherry-ATG8a. GFP-marked clones expressing RNAi or protein in the larval fat body were generated through heat shock-independent induction as previously described in ref. [60]. Fat bodies from both sexes were included.

**Immunofluorescence assays.** Dissected fat bodies were fixed in a solution of 4% PFA/PBS for 40 min. After permeabilization with 0.3% Triton/PBS, fat bodies were washed, and incubated overnight with anti-HA antibodies, and visualized using anti-mouse Alexa-633 (Invitrogen); antibody dilutions as indicated above. S2R + cells expressing Flag-dwg were fixed with 4% paraformaldehyde, permeabilized with 0.1% triton, and processed for immunostaining. DAPI (1 μg/ml) was used to stain nuclei. Samples were examined using a Zeiss LSM 780 confocal laser scanning microscope (Carl Zeiss Inc.) with a 63x Plan-Apochromat (NA1.4) objective lens.

### Co-Immunoprecipitation analysis in *Drosophila* cells

**Plasmids.** Full-length ORFs of *CG11486* (GEO01712), *CG9667* (GEO12785), *Deaf1* (GEO12259), *RfC4* (GEO04321), *CG7006* (GEO04456), *Larp* (GEO13890), *CG4813* (GEO05615), *lwr* (GEO03784), *DOR* (GEO05909), *dwg* (GEO06061), *wash* (GEO08420), *mri* (GEO13088), *me31B* (GEO01853), *MED15* (GEO05444), *NupS4* (GEO04647), *sm* (GEO09592), *smB* (GEO05072), *CG10209* (GEO04957), *MED4* (GEO04489), *Odj* (GEO06447), *Dpy-30L2* (GEO12106), *MED19* (GEO06036), *Chro* (GEO02531), *Atg8b* (GEO01803), *Atg8a* (GEO03266), *CG5446* (GEO09660), *CG4813* (GEO05615), *kin17* (GEO12682), *CG7484* (GEO09148), *sala* (GEO07724), and *CG9667* (GEO12785), from the FlyBi ORF clone collection reported in this work. ORFs were transferred into the *Drosophila* gateway vectors pAWF, pAGW or pAWG. The GFP ORF was cloned into pAWM as a control. To generate the Dwg deletion mutant proteins, DNA sequences corresponding to amino acids 1-150, 134-215, 215-393, and 385-592 of *dwg* were PCR amplified and subcloned into the pAWF vector. Using PCR mutagenesis, we generated *dwg^Y129A-I132A^*, *dwg^F401A-L404A^*, and *dwg^Y129A-I132A-F401A-L404A^* mutants by replacing tyrosine 129, isoleucine 132,

phenylalanine 401, or leucine 404 with alanine, followed by cloning into pAWF or pTWH. Mutant ORF sequences were verified by Sanger DNA sequencing.

**Antibodies.** Antibodies used for the study were as follows: anti-GFP (Molecular Probes, A6455), anti-Atg8 (Abcam, ab109364), anti-Flag (Sigma, F3165), and anti-GAPDH (GeneTex, GTX100118); antibody dilutions as indicated above.

**Cell culture.** *Drosophila* cells were cultured in Schneider's medium supplemented with 10% fetal bovine serum (FBS) at 25 °C. For Rapamycin (LC Laboratories, R-5000) or Bafilomycin (Sigma, B1793) treatment, S2R + cells were treated with 20 nM Rapamycin or 100 nM Bafilomycin-A1 (Baf-A1) for 24 h.

**Immunoprecipitation and immunoblotting.** DNA was transfected into S2R + cells in a 10 cm plate with Effectene transfection reagent (Qiagen) following manufacturer's protocol. After 3 days of incubation, cells were lysed using lysis buffer (Pierce) with protease inhibitor (Thermo Fisher Scientific) and phosphatase inhibitor (Sigma). Lysate was incubated with GFP-Trap agarose beads (Bulldog Bio) or anti-Flag M2 magnetic beads (Sigma) for 2 h at 4 °C to precipitate the protein complexes. Beads were washed 3–4 times with 1 ml lysis buffer. SDS-sample buffer was added, and the samples were boiled at 95 °C for 10 min. Boiled samples were run on polyacrylamide gel (Bio-Rad) and transferred to Immobilon-P polyvinylidene fluoride (PVDF) membrane (Millipore). The blot was probed with primary antibody, followed by HRP-conjugated secondary antibody, and signal was detected by enhanced chemiluminescence (ECL; Amersham).

**Quantification of mRNA expression.** Total RNA was extracted from control or *dwg^8^* mutants using TRIzol® reagent (Invitrogen). We synthesized the first strand cDNA with 1 μg of total RNA using iScript™ Reverse Transcription Supermix (BIO-RAD) followed by quantitative PCR with CFX96 Real-Time System (BIO-RAD) using iQ™ SYBR Green Supermix (BIO-RAD). All expression values were normalized to *RpL32* (also known as *rp49*). All assays were performed in triplicate. The primer sequences used for PCR are as follows:

Rp49: ATCGGTTACGGATCGAACAA, GACAATCTCCTTGCGCTTCT
Atg1: CTAAAGCCGTCGTCCAATGT, GAACAGCATGCTCCGGTATT
Atg17: GAAGCTGCACAACATCCCG, GCTGAGTAGCACGACACTTGG
Atg3: CCAAGACAAAACCCTACCTACC, GCCGACGTATTCCATCTGCT
Atg13: GAACCTAAAGACAGGAGAGAGCA, ACCCTCAGTCGTTTTCAGGGA

### Chromatin immunoprecipitation (ChIP)

S2R + cells expressing Flag-dwg were subjected to ChIP assays using SimpleChIP Plus Enzymatic Chromatin IP Kit (Cell Signaling Technology, 9005) according to the manufacturer's protocol. DNA co-immunoprecipitation with either anti-Flag antibody (Sigma, F3165) or IgG control antibody (a component of the kit) was analyzed by deep DNA-sequencing or quantified by ChIP-qPCR using primers shown below.

Atg1: CACTTGCAGGATCGATGGCA, TTACGCTGATCGTCCGTGTG
Atg17 promoter: CACATGCTCGGCCTGCTATT, CAGACTGTCGCTGGTGCTTT
Atg17 intron: TGCCCGCATCGTGTAAATGG, CTGCTGCTGCTGTGAGTGTT
Atg3: AGCTGCGAAGTGCAAGTCAA, GCGTTCAGATATTCGGCCACA
Atg13: AATCGCAGTGAAAGGGCGTT, AGTTCGCTGTCTGCGTTTGT

For ChIP-seq data analysis, low-quality reads and adaptor primer sequences were trimmed using Trim Galore 0.6.4 (https://github.com/FelixKrueger/TrimGalore), and trimmed reads were mapped against fly genome dm6 by bowtie2 2.3.5.1 with the additional argument "-q –local"[84]. Samtools 1.6 was used to sort, filter unique reads, and convert

file format to bam files[85]. Peak calling was performed with MACS2 2.2.6 using the additional parameter "-B –SPMR -f BAMPE -g dm"[86]. Peaks were annotated with HOMER 4.11[87]. DeepTools 3.4.0 were used for normalizing read counts to CPM and convert bam files to lyby format[88].

## Reporting summary

Further information on research design is available in the Nature Portfolio Reporting Summary linked to this article.

## Data availability

FlyBi binary interaction data and all data described in this study are provided without restrictions. These data are provided as Supplementary file 5 and are also available as a table and as a downloadable data file at the FlyBi project webpage ([https://flybi.hms.harvard.edu/]). In addition, these data have been integrated with other datasets at IntAct ([https://www.ebi.ac.uk/intact/])[44] and in the Molecular Interaction Search Tool (MIST; [https://fgrtools.hms.harvard.edu/MIST/])[33]. MAPPIT data is provided as Supplementary Data 6. RNAi data for the autophagy-related network is provided as Supplementary Data 8. Plasmid clones and associated information are available from both the *Drosophila* Genomics Resource Center (University of Indiana, Bloomington, IN) and the DNASU plasmid repository (Arizona State University, Phoenix, AZ). ORFs in the Gateway donor vector were end-read sequenced (see above, "Generation of a large-scale ORF clone resource"). Sequence data is available at GenBank and at the FlyBi project website (see Genbank Accession columns in the table at [https://flybi.hms.harvard.edu/results.php]). For a subset of 954 ORFs, the end-reads sequence spanned the full ORF. This sequence data is available at NCBI (Project Accession ID PRJNA349744) and a list of these ORFs, along with NCBI IDs, is available at the FlyBi project website (see [https://flybi.hms.harvard.edu/clones.php]). Interaction data was deposited at EBI IntAct (all *Drosophila* PPIs viewable at [https://www.ebi.ac.uk/intact/query/pubid:IM-28761]) and DroRI PPIs are available at MIST (see DroRI tab at [https://fgrtools.hms.harvard.edu/MIST/]). ChIPseq data is available at NCBI GEO (Accession ID GSE220887). Source data are provided with this paper.

## Code availability

The code for node shuffling used to generate random networks based on the FlyBi network is available at <https://github.com/moontreegy/flybi-network-analysis>. All other code was previously published. The L3 prediction code, together with example datasets, input data files and predictions, is available at <https://doi.org/10.5281/zenodo.2008592> (see ref. 43). Code relevant to Y2H and PPI analyses is available at <https://github.com/CCSB-DFCI/HuRI_paper> (see ref. 3). SAFE analysis software from Baryshnikova and colleagues is available at <https://github.com/baryshnikova-lab/safepy> (see ref. 72). FBgn IDs were updated and validated using <http://flybase.org/convert/id>[75].

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

## Acknowledgements

We thank Jonathan Zirin for helpful comments. This work was supported by NIH NHGRI 5R01HG007118 and NIH NIAMS 5R01AR057352 (to N.P.). Additional support includes NMRC Open Fund Individual Research (OF-IRG) Grant (OFIRG22jul-0007, to H.-W.T.) and a Foundation Grant from the Canadian Institutes of Health Research (to F.P.R.). M.V. is a Chercheur Qualifié Honoraire from the Fonds de la Recherche Scientifique (FRS-FNRS, Wallonia-Brussels Federation, Belgium). N.P. is an investigator of the Howard Hughes Medical Institute. This article is subject to HHMI's Open Access to Publications policy. HHMI lab heads have previously granted a nonexclusive CC BY 4.0 license to the public and a sub-licensable license to HHMI in their research articles. Pursuant to those licenses, the author-accepted manuscript of this article can be made freely available under a CC BY 4.0 license immediately upon publication.

## Author contributions

Contributed to overall project design and interpretation: H.-W.T., K.S., Y.H., T.H., M.A.C., D.E.H., S.E.C., M.V., N.P., S.E.M. Performed or supervised experiments: H.-W.T., K.S., R.B., D.Y.-Z., D.B., A.G.C., A.D., K.Y.G., D.E.H., C.L., J.J.K., I.L., M.L., W.X.L., D.M., C.P., S.R., S.D.R., S.S., B.T., K.H.W., F.P.R., M.A.C., S.E.C., M.V., N.P., S.E.M.; Performed or supervised data analysis, computation, integration, or interpretation: H.-W.T., K.S., Y.H., T.H., I.A.K., Y.G., D.Y.-Z., B.W.B., J.S.B., W.B., I.L., Y.L., K.L., R.L., J.R., Y.S., D.E.H., S.E.C., D.S., F.P.R., M.V., N.P., S.E.M.; Generated data visualizations: Y.H., T.H., Y.G., J.R., D.-K.K., H.J.L., D.S., M.A.C.; Contributed to design or methods development for a project component: H.-W.T., K.S., Y.H., T.H., J.J.K., I.L., F.P.R., M.A.C., J.T., D.E.H., S.E.C., M.V., N.P., S.E.M.; Submitted data or code to a repository: Y.H., T.H., S.E.C.; Wrote or edited the manuscript: H.-W.T., K.S., Y.H., T.H., D.-K.K., H.J.L., I.L., F.P.R., M.A.C., I.A.K., D.E.H., S.E.C., D.S., M.V., N.P., S.E.M.

## Competing interests

The authors declare no competing interests.

## Additional information

[1]Department of Genetics, Blavatnik Institute, Harvard Medical School, 77 Avenue Louis Pasteur, Boston, MA 02115, USA. [2]Program in Cancer and Stem Cell Biology, Duke-NUS Medical School, 8 College Road, Singapore 169857, Singapore. [3]Division of Cellular & Molecular Research, Humphrey Oei Institute of Cancer Research, National Cancer Centre Singapore, Singapore 169610, Singapore. [4]Center for Cancer Systems Biology (CCSB), Dana-Farber Cancer Institute, 450 Brookline Avenue, Boston, MA 02215, USA. [5]Department of Physics and Astronomy, Northwestern University, 633 Clark Street, Evanston, IL 60208, USA. [6]Northwestern Institute on Complex Systems, Chambers Hall, Northwestern University, 600 Foster St, Evanston, IL 60208, USA. [7]Howard Hughes Medical Institute, 77 Avenue Louis Pasteur, Boston, MA 02115, USA. [8]Berkeley Drosophila Genome Project, Lawrence Berkeley National Laboratory, 1 Cyclotron Rd, Berkeley, CA 94720, USA. [9]Department of Biomedical Engineering, Whiting School of Engineering, Johns Hopkins University, 3400 North Charles Street, Baltimore, MD 21218, USA. [10]High-Throughput Biology Center, Institute of Basic Biological Sciences, Johns Hopkins School of Medicine, 733 North Broadway, Baltimore, MD 21205, USA. [11]Donnelly Centre for Cellular and Biomolecular Research and Department of Molecular Genetics, University of Toronto, 160 College St, Toronto, ON M5S 3E1, Canada. [12]Lunenfeld-Tanenbaum Research Institute (LTRI), Sinai Health, 600 University Ave, Toronto, ON M5G 1×5, Canada. [13]Cytokine Receptor Lab, VIB Center for Medical Biotechnology, Albert Baertsoenkaai 3, 9000 Ghent, Belgium. [14]Department of Cancer Genetics and Genomics, Roswell Park Comprehensive Cancer Center, 665 Elm St., Buffalo, NY 14203, USA. [15]Department of Computer Science, University of Toronto, 40 St George St, Toronto, ON M5S 2E4, Canada. [16]These authors contributed equally: Hong-Wen Tang, Kerstin Spirohn. ✉e-mail: celniker@fruitfly.org; marc_vidal@dfci.harvard.edu; perrimon@genetics.med.harvard.edu; stephanie_mohr@hms.harvard.edu

