## [Peer Review File · Nature Communications]

Next-generation large-scale binary protein interaction network for *Drosophila melanogaster*REVIEWER COMMENTS

Reviewer #1 (Remarks to the Author):

Report on Tang et al. „Next-generation large-scale binary protein interaction network for Drosophila“

Tang et al. report a binary PPI data set for *Drosophila melanogaster* applying all-by-all Y2H screens of 10,000 *Drosophila* proteins: this is the 'FlyBi' dataset of 8,700 PPIs among some 3000 proteins. Orthogonal testing demonstrates the high quality of the data and overlap analysis using empirical statistics (Figure 2) is used for benchmarking. Then the data are expanded with quality filter (Giot et al at 0.7score) literature knowledge to 17000 interactions among 6500 proteins. The analyses performed follow largely approaches from human PPI mapping and are straight forward. In the "second" part the novel resource is used to generate an autophagy interaction network and identify new PPIs with four known autophagy genes (e.g. Atg8a). In summary, 116 novel candidates that we validated in vivo using two different autophagy-related assays. First, loss-of-function experiments using RNAi in GMR-Gal4>UAS-Atg1 lines with overexpression of Atg1, where high level of autophagy, gives a rough eye phenotype: 137/166 genes were tested and perturbed the phenotype together with a background gene set. (~ 50% ppi targets, background modifier control set 24%). In an ATG8 redistribution assay 60 (26%) increased fat body Atg8a puncta (inhibitors of autophagy), while 41 ko lines (18%) inhibited (activators of autophagy), finally 55% are positive in IP from SR cells. Functionally, they focus in on dwg, a chromatin binding insulator protein, that was identified as putative negative regulator of autophagy. It acts through binding near a set of ATG genes (via ChIP-seq) and suppresses their transcription (not shown). Mechanistically it works through direct interaction with ATG8a. The mostly nuclear dwg is a substrate of autophagy through interaction with ATG8 impacting its levels. Yes! a drosophila PPI network update is urgently needed and its advance, useful and potential impact is substantial. Specifically, many leads in the autophagy network can be further exploited and thus the data will also impact on the autophagy field.

Points for consideration (I actually have only minor points)

*) How exactly are the pairs predicted from the screen data: "The resulting list of sequence-confirmed putative binary interactors was supplemented by using a computational network-based approach to predict additional interaction pairs based on pairs in the assay version 1 screens followed by experimental validation (see Fig. 1A, Methods, and 37), altogether resulting in 332 experimentally predictions confirmed in the pairwise test. " What is the principle, it should be explained without the need to read 37. Or are the predictions simply taken from 37?

*) Figure 1A is not as clear as it could possibly be. It is not clear whether the 4 screens stem from repeating v1 and v3 each, or whether there are four different configurations used.

"two screens in each of two different configurations differing in the position of the Gal4 activation domain (AD) fusion, i.e. N- or C-terminus, and in the overall level of exogenous expression, i.e. using either centromeric or two-micron based expression vectors ... (Fig. 1A and Suppl. Fig. 1).

Screen_1, Cen X Cen,

Screen_2, Cen X Cen,

Screen_3, Cen X 2u,

Screen_4, Cen X 2u,

I think somehow, e.g. in Figure1 and in the suppl table the AD configuration (C or N) is missing.

*) Why is the configuration in Mappit chosen the same as in Y2H? Is there a difference in PPR or PRR testing with the C-terminal version of Mappit? Q refers to "Literature pairs also did not validate at the same rate when tested using the C-terminal version of Mappit. We attribute this to the fact that the MAPPIT assay has not been optimized for screens performed using C-terminally-fused ORFs". This is confusing because the difference in the results relates to the Y2H configuration i.e. the screens, but maybe due to the Mappit configuration. Or in other words, if Mappit N-C is the problem why not testing Y2H screen v3 and v4 pairs with Mappit N-N?

- *) Figure 2. "We generated 1,000 randomized versions of the FlyBi network by node shuffling." Was the degree of the nodes preserved in the shuffling procedure e.g. shuffling using binning of the nodes according to degree?
- *) Figure 3 B. The description in the legend does not match the figure, e.g. edge color etc. and the figure overall is not very informative.
- *) In Figure 3D a control image fed and starved of the wt-lines is missing for comparison of the effects of ATG8 localization. It contrasts two extremes, activation und feeding and inhibition under starving.
- *) Data in Figure 4. Actual suppression of ATG 1, 3 ,12 and 17 transcription or likewise upregulation upon dwg reduction is not shown or active chromatin marks.
- *) Figure 4G. The quality is poor (also in comparison to Figure 3D). While the colocalization of ATG8 and dwg is well visible in the images with wt overexpression (starved), the effects on autophagy can not be assessed from the images.
- *) Is ATG8 expression somehow regulated by dwg?
- *) Can the ATG8 binding mutations in dwg affect chromatin function e.g. in whatever the isolator recruits or binds?
- *) In mentioning that protein network mapping did not progress for two decades, the authors may have forgotten some of their own work ...
- i) Vinayagam et al. (2016) An Integrative Analysis of the InR/PI3K/Akt Network Identifies the Dynamic Response to Insulin Signaling.; ii) Kwon et al. (2013) The Hippo Signaling Pathway Interactome and ... I guess other studies as well.

Reviewer #2 (Remarks to the Author):

This is an important paper that identifies a substantial number (~8000) of high-quality *Drosophila* binary protein-protein interactions (PPI) using yeast two-hybrid screens. The authors call this new dataset FlyBI. The authors benchmark the quality of FlyBI and of other *Drosophila* PPI datasets by testing a subset of each using an orthogonal PPI assay, MAPPIT. Based on that metric, they show that FlyBI is of similar quality to binary PPI from the literature found in more than one publication or experiment (the so-called Lit-BM data), and better than a number of other datasets including PPI found only once in the literature. Based on that finding they combine FlyBI, Lit-BM, and other data benchmarked to be of similar high quality into a single dataset and call it the *Drosophila* reference set of PPI, DroRI, with about 17,000 PPI. The new data (FlyBi) and the newly benchmarked other data (in DroRI) will be a valuable resource for the research community and the authors are making it available in the increasing useful database, MIST. The paper also demonstrates the value of the PPI by genetically characterizing a network of proteins involved in autophagy, providing some very interesting insights into autophagy. Combined, these are significant advancements worthy of publication in Nat Comm.

The article could be improved by addressing a few minor issues. The most important issue to address is to make all of the data discussed in the paper available and properly annotated. Examples:

- The PPI listed in the Supp tables should include the source of the data, publication(s), database(s), and methods, like the data found in MIST (in most cases this could be easily achieved by downloading the relevant subset of data from MIST and putting it into the Supp tables). This includes the PRS in Supp table 1, the PPI in Supp tables 2, 3, 7, 8, and even the FlyBI data in Supp table 5; so that we know which of the FlyBI interactions were also detected in previous data (this is important information that is not otherwise given in the paper).
- The authors tested PPI from various datasets in MAPPIT (Fig 1B, Supp Fig 2, Supp Fig 3). The Supp tables should list the interactions tested, their source (as noted above), and whether they were positive or not. This could be done with one table or multiple tables but should include PPIs tested from FlyBI, PRS, RRS, CuraGenH and L, Lit-BM, Lit-BS?, Finley(?), DPIM, other Y2H data, etc.; i.e., everything in the figures This is critical primary data that must be included in the paper. Otherwise, all

we are presented with are fractions positive, without any indication of how many and which interactions were tested for each dataset.

- Just to illustrate how this data is not currently available: All of the PPI in Table 6 (Y2H tested in MAPPIT) are from Fly-BI (Table 5) only (none from the other datasets tested). About half of the MAPPIT positives listed in Table 8 are from FlyBI (Table 5) and the other half are from origins unknown; none of them appear to be from the Lit-BM tested interactions.

Other comments

Line 121: The limitations of the Giot 2003 PPI map do not include that the baits and preys were not sequenced or were unknown. Thus, these statements starting on line 121 should be modified (or removed): "A significant limitation of that study was that at the time it was not feasible to sequence the full collection, such that the identities of all bait-prey pairs tested are not known. In addition, which partial or full-length isoforms were tested is not known...". In fact, as indicated in Figure 1 of Giot 2003, the identities of 10,623 individual bait clones and 10,878 individual prey clones were verified (to be the intended full-length clone) by sizing and sequencing before the screen began. Moreover, during the screen, individual sequence-verified bait clones were used and the identities of all interacting prey proteins were determined by sequencing. Acknowledging this does not detract from the author's accomplishment of developing a new, much more useful clone set.

Starting on line 133: It would be useful to give more numbers here. How many FiPP were detected, how many of those were retested in pair-wise assays, and how many were positive in the retests?

Line 134-137: This begins a confusing discussion of the computational predictions and testing. First, how does Fig 1A illustrate any of that process? Further, how many new interactions were predicted in the network-based computational approach, how many of those were tested in pair-wise assays (and what were those assays), and how many were positive in the pair-wise assay? As written, it appears that just 332 PPI were predicted, tested, and confirmed. Yet later (starting line 144), and out of place, they give the pairwise testing positive rates for the computational predictions (presumably at different confidence cut offs). These facts should go with a better description of the computational approach starting at line 135.

Starting at line 184: Were any of the Lit-BM PPI allowed in the random sets of DPIM and other Y2H PPI selected for MAPPIT testing? If not, then those random sets were not random; they excluded predetermined higher confidence PPI. This is one of the many reasons to include all of the data as noted above.

Line 705 (legend to Fig 1C). "...the total number of binary interactions accumulated in the literature (Lit-BM), as curated in FlyBase and displayed based on date of publication.." Do the literature interactions include only those curated by Flybase or also those curated by other databases – like IntAct and BioGRID? At times, the IntAct and BioGRID had a large curated set of Drosophila PPI that were not available in Flybase. It is OK to be using just the Flybase-curated set of literature interactions for this study, but it is not OK to imply that those were all that the literature had to offer, as Figs 1C and 2B is doing.

Fig 1B, Supp Fig 2, and Supp Fig 3 show the % MAPPIT positive for various data sets. The text or legend should indicate the number tested in each case. Moreover, the Supp tables should list the interactions tested, their source, and whether they were positive or not, as noted above.

Fig 1B and Supp Figs 2 and 3: How were the error bars calculated in Fig 1B and Supp Figs 2 and 3? The figs show a ratio (%) of the number of MAPPIT positive to the number tested. Where is the uncertainty in those two numbers? The error bars are mentioned only in the legend to Supp Fig 3, which says they indicate "standard of error." That does not explain how they were calculated or why

they are there at all.

Line 252, 253: The DOR-Atg8a interaction had also been detected long before by the Curagen screen and by coimmunoprecipitation in another paper. This illustrates that some FlyBI PPI had already been detected. This does not detract from the FlyBI data, but rather adds value to it. Again, it would be useful to annotate the FlyBI interactions to indicate which ones had been identified, how, and by what paper (as done in MIST).

Lines 275 and 289 and 391: A good idea, but not novel, right? Using PPI data to guide a more efficient knockdown screen has been done, probably several times, including one with *Drosophila* that used mostly Y2H data (Guest 2011, PMID: 21548953).

Line 361: It says that DroRI has 8,723 PPI, probably meaning 17,232.

Line 262-364: "Features that distinguish the FlyBi project from past efforts..." Computational predictions of PPI have been done in many past projects. The author's version of this approach was excellent and should be highlighted, but the general idea is not a feature that distinguishes from past projects. In one example, Schwartz 2011 (PMID: 19079254) used two different computational approaches to predict *Drosophila* PPI and then used two versions of the two-hybrid system to test them and found 450 new high confidence PPI.

A few comments about how this data is presented in MIST (which could be fixed after this paper is published):

The DroRI data in MIST is listed without attribution. According to the authors, about half of these are from FlyBI and the other half are from Lit-BM. However, the source database is just listed as "extend" and many lack any PubMedID, even many that are clearly from the literature.

Clearly, the MIST people should be working to integrate the DroRI data with the rest of the *Drosophila* data in MIST. For example, rather than giving DroRI it's own tab, the data should be in with the rest of the fly PPI, but with annotations indicating that it is part of DroRI, and found by FlyBI where appropriate. In the meantime, data in DroRI tab should be fully annotated.

A related issue is that MIST site does not explain how the "Ranks" are determined. I found at least one PPI that is ranked "low" in the fly dataset, but high in the DroRI.

Reviewer #3 (Remarks to the Author):

The manuscript by Tang et al presents a new and greatly expanded binary interactome for *Drosophila*, generated by the yeast two-hybrid system supplemented with computationally predicted (and then validated) interactions. I fully agree with the authors that binary interactomes are highly valuable for the scientific community, and that improvements in approaches and analyses make it worthwhile to continue to invest in interactomes for widely-used model systems, like *Drosophila*.

The experiments have been done well by absolute experts in the field. The experimental and computational pipeline has been developed over the past two decades, and was recently applied to generate a human interactome map. Here, the same pipeline is applied to *Drosophila* proteins. Starting from validated ORF clones, a highly optimized two-hybrid framework is used that is followed up by quality validation using the orthogonal MAPPIT assay, and computational comparisons with other datasets. The resulting network should be of the highest quality that is currently attainable, and this is evidenced by comparison with high-quality literature derived data. FlyBi will undoubtedly be

used by many *Drosophila* researchers and I highly support its publication. The manuscript ends with an illustrative example where the network is used to gain new insight into autophagy, ultimately focusing on the regulator deformed wings.

I have no experimental suggestions but the clarity of the manuscript could be improved in certain sections.

I found the logic of the writing in the section lines 138-150 confusing. It almost seems like the order is not correct or a section of text is missing. In line 137 the description of the pipeline up to and including the retesting and L3 additions are described. I would expect this to be followed by the text starting on line 151 (the numbers that were identified in the pipeline). Instead, on line 138 the authors describe some sort of initial quality test. This starts by establishing 4 sets of protein pairs, but then no quality assessment data is presented for these 4 sets. Instead, the authors describe pairwise testing results for the L3 predictions, without mentioning what kind of pairwise test was performed. These data are only described in text, with no accompanying figure. I also don't understand how 71% of 1000 predictions tested positive, yet only 332 computationally predicted pairs are considered confirmed and added to FlyBi. Finally, how was the PRS set established? Was this a random selection of FlyBi (that would be ideal for a quality control dataset), or a manually selected set of pairs? Can the authors clarify this section?

On line 206 the authors conclude that the level of enrichment in GO and/or phenotype annotations is comparable between FlyBi and Lit-BM-20. Yet in Figure 2C the enrichment odds values for Lit-BM-20 are higher than for FlyBi, particularly since a log scale was used. Why is the conclusion that the levels of enrichment are similar between the two datasets? Can the authors apply statistics to prove this?

I was surprised by the low overlap between FlyBi and DPiM. I know many arguments why this may be the case, and the authors briefly mention some. Yet I was expecting that as datasets become more complete and more accurate, the overlap would start to increase. Would it be possible to expand the discussion beyond 'experimental and biological differences'? For instance, within a protein complex binary interactions will occur. Not all of these will be able to form between the two proteins in isolation, but intuitively I expect that some fraction will. Is there any literature or analysis that might support that the majority of protein interactions that take place within a protein complex cannot be reconstituted between the two proteins in isolation? Or is there evidence that we have still only scratched the surface of interactions and we would not expect to find overlap?

Quite a high fraction of genes tested modify the autophagy eye phenotype of Atg1 overexpressing flies. The authors nicely perform a control experiment to show that this is not (or to a lesser extent) the case for a random selection of genes. In the larval Fat body, again a large fraction of genes modified the Atg8a puncta formation (positively or negatively). A comparison between assay (overlap in results) would be informative.

Line 300: a 55% retest rate by IP is in line with previous retest rates of high quality datasets I believe. I would be useful to indicate if this is the case.

Line 507: are positive pairs picked into individual wells? It wasn't clear to me if the PCR analysis in this step does not have to deal with the potential of multiple interactions in one well (I assume so)

Line 509 refers to Luck et al for computational analysis to generate a list of binary interactors identified in the screens. It wasn't clear to me what analysis this refers to. In the retest section of the referred paper the most relevant I could find was 'Only pairs identified from the pipeline that matched the tested ORF pairs previously scored as positive were considered PPIs.', but that is likely not what is meant by computational analysis.

Line 136 refers to Fig. 1A to support the supplementing of the network with validated computational

predictions, but there is no data/drawing regarding computational predictions in this panel.

Line 137: experimentally predictions should presumably read experimentally validated computational predictions

REVIEWER COMMENTS

Reviewer #1 (Remarks to the Author):

Report on Tang et al. „Next-generation large-scale binary protein interaction network for *Drosophila*”

Tang et al. report a binary PPI data set for *Drosophila melanogaster* applying all-by-all Y2H screens of 10,000 *Drosophila* proteins: this is the ‘FlyBi’ dataset of 8,700 PPIs among some 3000 proteins. Orthogonal testing demonstrates the high quality of the data and overlap analysis using empirical statistics (Figure 2) is used for benchmarking. Then the data are expanded with quality filter (Giot et al at 0.7score) literature knowledge to 17000 interactions among 6500 proteins. The analyses performed follow largely approaches from human PPI mapping and are straight forward.

In the “second” part the novel resource is used to generate an autophagy interaction network and identify new PPIs with four known autophagy genes (e.g. Atg8a). In summary, 116 novel candidates that we validated in vivo using two different autophagy-related assays. First, loss-of-function experiments using RNAi in GMR-Gal4>UAS-Atg1 lines with overexpression of Atg1, where high level of autophagy, gives a rough eye phenotype: 137/166 genes were tested and perturbed the phenotype together with a background gene set. (~ 50% ppi targets, background modifier control set 24%). In an ATG8 redistribution assay 60 (26%) increased fat body Atg8a puncta (inhibitors of autophagy), while 41 ko lines (18%) inhibited (activators of autophagy), finally 55% are positive in IP from SR cells. Functionally, they focus in on dwg, a chromatin binding insulator protein, that was identified as putative negative regulator of autophagy. It acts through binding near a set of ATG genes (via ChIP-seq) and suppresses their transcription (not shown). Mechanistically it works through direct interaction with ATG8a. The mostly nuclear dwg is a substrate of autophagy through interaction with ATG8 impacting its levels.

Yes! a *drosophila* PPI network update is urgently needed and its advance, useful and potential impact is substantial. Specifically, many leads in the autophagy network can be further exploited and thus the data will also impact on the autophagy field.

Points for consideration (I actually have only minor points)

R1-1) How exactly are the pairs predicted from the screen data: “The resulting list of sequence-confirmed putative binary interactors was supplemented by using a computational network-based approach to predict additional interaction pairs based on pairs in the assay version 1 screens followed by experimental validation (see Fig. 1A, Methods, and 37), altogether resulting in 332 experimentally predictions confirmed in the pairwise test. “ What is the principle, it should be explained without the need to read 37. Or are the predictions simply taken from 37?

We appreciate the comment. In response to both this comment and a similar comment from another reviewer, we revised the section that describes the computational approach and results.

The revised paragraph now begins: “The list of sequence-confirmed putative binary interactors resulting from experimental Y2H testing was supplemented by application of a computational approach to predict additional interaction pairs based on pairs identified in screens 1 and 2 (assay version 1). Different from other approaches previously applied to *Drosophila* (e.g., see 42), our method, known as the L3 approach, is based on connectedness of proteins within a network (see Methods and 43). Following application of the L3 approach ...”

R1-2) Figure 1A is not as clear as it could possibly be. It is not clear whether the 4 screens stem from repeating v1 and v3 each, or whether there are four different configurations used.

“two screens in each of two different configurations differing in the position of the Gal4 activation domain (AD) fusion, i.e. N- or C-terminus, and in the overall level of exogenous expression, i.e. using either centromeric or two-micron based expression vectors ... (Fig. 1A and Suppl. Fig. 1).

Screen_1, Cen X Cen,

Screen_2, Cen X Cen,

Screen_3, Cen X 2u,

Screen_4, Cen X 2u,

I think somehow, e.g. in Figure1 and in the suppl table the AD configuration (C or N) is missing.

We thank the reviewer for raising this concern. For clarity, we (a) changed the shading on two of the four boxes in Fig. 1A to visually indicate that two of the screens were in one format and two were in the other format, (b) made a corresponding change to the shading of the version 2 AD fusion also in Fig. 1A, and (c) updated Supplemental Table 11 with clarification information regarding the assay versions, i.e., changed the Table 11 column header “pDEST-DB” to “pDEST-DB (used for both assay versions),” changed “pDEST-AD-CHY2” to “pDEST-AD-CHY2 (used for assay version 1),” and changed “pDEST-AD-AR68” to “pDEST-AD-AR68 (used for assay version 3).”

R1-3) Why is the configuration in Mappit chosen the same as in Y2H? Is there a difference in PPR or PRR testing with the C-terminal version of Mappit? Q refers to “Literature pairs also did not validate at the same rate when tested using the C-terminal version of Mappit. We attribute this to the fact that the MAPPIT assay has not been optimized for screens performed using C-terminally-fused ORFs”. This is confusing because the difference in the results relates to the Y2H configuration i.e. the screens, but maybe due to the Mappit configuration. Or in other words, if Mappit N-C is the problem why not testing Y2H screen v3 and v4 pairs with Mappit N-N?

We appreciate the questions raised. Y2H screens 3 & 4 were done using an N-C configuration, i.e., N terminal DB and C-terminal AD. We have observed in general for Y2H studies that even when tested using the same assay, different fusion configurations result in detection of different pairs, both in systematic screens and when testing a PRS, which we also reported in Luck et al. For FlyBi, when N-C Y2H pairs and literature N-C pairs were tested using N-N MAPPIT, detectability was less than what was seen when N-N Y2H pairs and N-N literature pairs were tested using N-N MAPPIT. Although lower than literature N-N pairs, detectability of both sets was still above the 1% RRS threshold. N-C Y2H pairs and N-C literature pairs were more

detectable by N-C MAPPIT, again above the RRS threshold although N-C PRS detectability was lower than for N-N PRS pairs. In summary, although using MAPPIT in N-C configuration is not ideal because the assay itself is optimized for N-N studies, using MAPPIT in N-C configuration nevertheless performs better for testing of pairs identified in the N-C configuration Y2H assays as compared with using MAPPIT in N-N configuration.

To help clarify this point for readers, we made significant modifications to the paragraph that begins “The results of this analysis made it possible for us to apply a cut-off value for CuraGen pairs that produced a list of pairs of equivalent high quality as compared with FlyBi pairs from assay version 1 ...”.

R1-4) Figure 2. “We generated 1,000 randomized versions of the FlyBi network by node shuffling.” Was the degree of the nodes preserved in the shuffling procedure e.g. shuffling using binning of the nodes according to degree?

No, the degree of the nodes was not necessarily preserved in our node shuffling procedure. We added a note to indicate this in Methods, subsection Bioinformatic analyses, sub-subsection Gene set enrichment.

R1-5) Figure 3 B. The description in the legend does not match the figure, e.g. edge color etc. and the figure overall is not very informative.

We apologize for the error and agree that the figure overall is not very informative. We removed the original panel B from our revised Figure 3.

R1-6) In Figure 3D a control image fed and starved of the wt-lines is missing for comparison of the effects of ATG8 localization. It contrasts two extremes, activation und feeding and inhibition under starving.

We appreciate the concern but feel that adding Atg8 localization in wildtype under fed and starved conditions is not needed and would take up precious space in the figure. The tagged reagent and the redistribution under fed and starved conditions is well established (e.g., see Nezis et al. 2009 PMID: 19066465; Jacomin and Nezis 2016 PMID: 27557573; Tang et al. 2018 PMID: 29606597, and Tang et al. 2021 PMID: 33649236).

R1-7) Data in Figure 4. Actual suppression of ATG 1, 3, 12 and 17 transcription or likewise upregulation upon *dwg* reduction is not shown or active chromatin marks.

We thank this reviewer for the comment. Our ChIP-seq and ChIP-qPCR both suggest that *Dwg* can bind to promoter regions of the *Atg1*, *Atg3*, *Atg13*, and *Atg17* genes (Figure 4C-D). In *dwg* mutants, mRNA levels of these four Atg genes are increased (Figure S7), strongly suggesting that *dwg* suppresses autophagy by inhibiting Atg gene expression. We recognize that additional

experiments that could be done to further support the conclusion. However, we feel that this type of in-depth study is beyond the scope of the current work.

R1-8) Figure 4G. The quality is poor (also in comparison to Figure 3D). While the colocalization of ATG8 and *dwg* is well visible in the images with *wt* overexpression (starved), the effects on autophagy can not be assessed from the images.

We replaced one set of images, changed the size of the images, and improved the resolution of the figure file to address this concern. We will furthermore work with the journal to make sure that the resolution of the final images maintains a high standard.

R1-9) Is ATG8 expression somehow regulated by *dwg*?

We appreciate the question being raised. Our qPCR and immunoblot results suggest that the *Atg8a* mRNA and protein levels are increased in *dwg* mutant animals. This is suggestive of the idea that ATG8 expression might be regulated by *Dwg*. However, we feel that fully exploring this idea will take additional detailed analyses and is outside the scope of this report.

R1-10) Can the ATG8 binding mutations in *dwg* affect chromatin function e.g. in whatever the isolator recruits or binds?

We appreciate your raising this question. We showed that depletion of *Dwg* induces autophagy through up-regulation of autophagy-related transcripts (Fig. 3D). In contrast, overexpression of *Dwg*^{4A}, a *dwg* form with mutations in the putative ATG8 binding sites, can suppress starvation-induced autophagy. This suggests that *Dwg*^{4A} still can bind to chromatin and inhibit autophagy. Future work is needed to further test if *Dwg*^{4A} can associate with DNA or change its interacting proteins. However, we feel that these additional assays are beyond the scope of the current study.

R1-11) In mentioning that protein network mapping did not progress for two decades, the authors may have forgotten some of their own work ... i) Vinayagam et al. (2016) An Integrative Analysis of the InR/PI3K/Akt Network Identifies the Dynamic Response to Insulin Signaling.; ii) Kwon et al. (2013) The Hippo Signaling Pathway Interactome and ... I guess other studies as well.

We appreciate the reviewer pointing this out. In response, we revised the previous statement early in the main text “Particularly relevant to this study, since the release of the last binary PPI map for *Drosophila melanogaster* two decades ago, methods for identification of binary interactions have improved and caveats to the approach are now well understood.”

The revised statement now reads “Particularly relevant to this study, methods for identification of binary interactions have improved over the years and caveats to the approach are now well understood.”

We also revised the text in the paragraph that begins “Protein-based resources and datasets provide an important complement ...” to include mention of the studies mentioned by the reviewer and other examples of smaller-scale studies.

Reviewer #2 (Remarks to the Author):

This is an important paper that identifies a substantial number (~8000) of high-quality *Drosophila* binary protein-protein interactions (PPI) using yeast two-hybrid screens. The authors call this new dataset FlyBI. The authors benchmark the quality of FlyBI and of other *Drosophila* PPI datasets by testing a subset of each using an orthogonal PPI assay, MAPPIT. Based on that metric, they show that FlyBI is of similar quality to binary PPI from the literature found in more than one publication or experiment (the so-called Lit-BM data), and better than a number of other datasets including PPI found only once in the literature. Based on that finding they combine FlyBI, Lit-BM, and other data benchmarked to be of similar high quality into a single dataset and call it the *Drosophila* reference set of PPI, DroRI, with about 17,000 PPI. The new data (FlyBi) and the newly benchmarked other data (in DroRI) will be a valuable resource for the research community and the authors are making it available in the increasing useful database, MIST. The paper also demonstrates the value of the PPI by genetically characterizing a network of proteins involved in autophagy, providing some very interesting insights into autophagy. Combined, these are significant advancements worthy of publication in Nat Comm.

The article could be improved by addressing a few minor issues. The most important issue to address is to make all of the data discussed in the paper available and properly annotated. Examples:

R2-1) The PPI listed in the Supp tables should include the source of the data, publication(s), database(s), and methods, like the data found in MIST (in most cases this could be easily achieved by downloading the relevant subset of data from MIST and putting it into the Supp tables). This includes the PRS in Supp table 1, the PPI in Supp tables 2, 3, 7, 8, and even the FlyBI data in Supp table 5; so that we know which of the FlyBI interactions were also detected in previous data (this is important information that is not otherwise given in the paper).

We updated the tables in accordance with this suggestion. The tables now include information from MIST about the source databases, publications, and methods.

R2-2) The authors tested PPI from various datasets in MAPPIT (Fig 1B, Supp Fig 2, Supp Fig 3). The Supp tables should list the interactions tested, their source (as noted above), and whether they were positive or not. This could be done with one table or multiple tables but should include PPIs tested from FlyBI, PRS, RRS, CuraGenH and L, Lit-BM, Lit-BS?, Finley(?), DPIM, other Y2H data, etc.; i.e., everything in the figures This is critical primary data that must be included in the paper. Otherwise, all we are presented with are fractions positive, without any indication of how many and which interactions were tested for each dataset.

We put all information related to MAPPIT testing into one supplemental table (new Suppl. Table 6) that replaces what had appeared as Suppl. Tables 6, 7 and 8 in the previous version. The new table lists the gene pairs from all sources mentioned in addition presenting the results.

R2-3) Just to illustrate how this data is not currently available: All of the PPI in Table 6 (Y2H tested in MAPPIT) are from Fly-BI (Table 5) only (none from the other datasets tested). About half of the MAPPIT positives listed in Table 8 are from FlyBI (Table 5) and the other half are from origins unknown; none of them appear to be from the Lit-BM tested interactions.

We apologize for the lack of access to relevant information. We have updated the tables to address the concern, e.g., please see “category” in the new Suppl. Table 6.

Other comments

R2-4) Line 121: The limitations of the Giot 2003 PPI map do not include that the baits and preys were not sequenced or were unknown. Thus, these statements starting on line 121 should be modified (or removed): “A significant limitation of that study was that at the time it was not feasible to sequence the full collection, such that the identities of all bait-prey pairs tested are not known. In addition, which partial or full-length isoforms were tested is not known...”. In fact, as indicated in Figure 1 of Giot 2003, the identities of 10,623 individual bait clones and 10,878 individual prey clones were verified (to be the intended full-length clone) by sizing and sequencing before the screen began. Moreover, during the screen, individual sequence-verified bait clones were used and the identities of all interacting prey proteins were determined by sequencing. Acknowledging this does not detract from the author’s accomplishment of developing a new, much more useful clone set.

We edited these passages accordingly. The quoted portion now reads as follows: “Limitations of the study include that only a single assay version was used; a limited number of replicate screens were performed; and gene annotations were of poorer quality than they are now. In the study, a subset of 4,780 PPIs were reported as reaching acceptable quality levels.”

R2-5) Starting on line 133: It would be useful to give more numbers here. How many FiPP were detected, how many of those were retested in pair-wise assays, and how many were positive in the retests?

While we agree with the reviewer in principle that it is often useful to provide more numbers, in our experience we have found that the number of FiPPs and the retest rate are not useful and can even be misleading. Providing the actual number of colonies picked and identified by sequencing would be potentially useful. However, while we did sequence all colonies using Illumina Solexa technology to identify interacting partners, as part of our high-throughput pipeline, we typically do not determine precisely which colony produced the FiPP. Instead, we simply use the sequence data to identify ORF pairs, then individually select (‘cherry-pick’) query ORFs from the bait and prey collections to do lower throughput pairwise testing as a retest. Considering the noise associated with next-generation sequencing data, we also used “a cutoff

that balances the risk of testing too many false positives FiPPs versus not testing too many true positive FiPPs” (see Methods). Thus, the number of FiPPs detected is a function of the SWIM score cutoff. We tested all FiPPs above the cutoff value. The retest positive rate was also dependent on the cutoff.

R2-6) Line 134-137: This begins a confusing discussion of the computational predictions and testing. First, how does Fig 1A illustrate any of that process? Further, how many new interactions were predicted in the network-based computational approach, how many of those were tested in pair-wise assays (and what were those assays), and how many were positive in the pair-wise assay? As written, it appears that just 332 PPI were predicted, tested, and confirmed. Yet later (starting line 144), and out of place, they give the pairwise testing positive rates for the computational predictions (presumably at different confidence cut offs). These facts should go with a better description of the computational approach starting at line 135.

We thank the reviewer for raising these concerns. In response to this comment and a related comment from Reviewer 1, we have extensively revised the relevant section.

R2-7) Starting at line 184: Were any of the Lit-BM PPI allowed in the random sets of DPIM and other Y2H PPI selected for MAPPIT testing? If not, then those random sets were not random; they excluded predetermined higher confidence PPI. This is one of the many reasons to include all of the data as noted above.

We appreciate the concern but can confirm that we did not exclude high-confident PPIs from the random set.

R2-8) Line 705 (legend to Fig 1C). “..the total number of binary interactions accumulated in the literature (Lit-BM), as curated in FlyBase and displayed based on date of publication..” Do the literature interactions include only those curated by Flybase or also those curated by other databases – like IntAct and BioGRID? At times, the IntAct and BioGRID had a large curated set of Drosophila PPI that were not available in Flybase. It is OK to be using just the Flybase-curated set of literature interactions for this study, but it is not OK to imply that those were all that the literature had to offer, as Figs 1C and 2B is doing.

We appreciate the concern but can confirm that our “literature interactions” set includes not only pairs curated by Flybase but also pairs curated by other databases – like IntAct and BioGRID. We removed “as curated in FlyBase” from the figure legend text.

R2-9) Fig 1B, Supp Fig 2, and Supp Fig 3 show the % MAPPIT positive for various data sets. The text or legend should indicate the number tested in each case. Moreover, the Supp tables should list the interactions tested, their source, and whether they were positive or not, as noted above.

We modified the legend to include the number tested in each category. In addition, we updated the supplemental tables according to the suggestion (a new Suppl. Table 6 replaces the previous Suppl. Tables 6, 7, and 8).

R2-10) Fig 1B and Supp Figs 2 and 3: How were the error bars calculated in Fig 1B and Supp Figs 2 and 3? The figs show a ratio (%) of the number of MAPPIT positive to the number tested. Where is the uncertainty in those two numbers? The error bars are mentioned only in the legend to Supp Fig 3, which says they indicate “standard of error.” That does not explain how they were calculated or why they are there at all.

We appreciate the comment. We can confirm that there is no uncertainty in the number tested and the number of positives. We used the ratio of the sample from each dataset to estimate the ratio of the whole population. The error bar indicates the uncertainty or variation of the estimate. We changed the legend text to indicate “standard error of proportion.” We calculated this using the following formula:

$$SE = \sqrt{p(1-p)/n}$$

Where p is the ratio of the number of positive to the number tested and n is the number tested.

R2-11) Line 252, 253: The DOR-Atg8a interaction had also been detected long before by the Curagen screen and by coimmunoprecipitation in another paper. This illustrates that some FlyBi PPI had already been detected. This does not detract from the FlyBi data, but rather adds value to it. Again, it would be useful to annotate the FlyBi interactions to indicate which ones had been identified, how, and by what paper (as done in MIST).

We appreciate the comment. We updated the relevant section of the text to mention that the DOR-At8a interaction has previously been detected, including citation of the studies. We have also now annotated the FlyBi data as suggested (revised Suppl. Table 5, column I).

R2-12) Lines 275 and 289 and 391: A good idea, but not novel, right? Using PPI data to guide a more efficient knockdown screen has been done, probably several times, including one with *Drosophila* that used mostly Y2H data (Guest 2011, PMID: 21548953).

We appreciate this point being raised. We did not mean to imply that the overall approach was novel. Instead, our intent was to convey that the knockdown screen data are consistent with the idea that the FlyBi data are of high quality and have value to the community. We added the following statement at the original line 275: “This is consistent with a previous report that showed that protein network information can be used to limit false discovery in *Drosophila* RNAi screens (Guest et al. 2011).” We also changed “Importantly” to “Moreover, as expected for high-quality data ...” at the original line 391.

R2-13) Line 361: It says that DroRI has 8,723 PPI, probably meaning 17,232.

Thank you for catching this error. We inadvertently repeated the FlyBi numbers where we meant to provide the DroRI numbers. We corrected the error in the revised version.

R2-14) Line 262-364: “Features that distinguish the FlyBi project from past efforts...”
Computational predictions of PPI have been done in many past projects. The author’s version of this approach was excellent and should be highlighted, but the general idea is not a feature that distinguishes from past projects. In one example, Schwartz 2011 (PMID: 19079254) used two different computational approaches to predict Drosophila. PPI and then used two versions of the two-hybrid system to test them and found 450 new high confidence PPI.

We thank the reviewer for the correction. We modified the statement accordingly.

A few comments about how this data is presented in MIST (which could be fixed after this paper is published):

R2-15) The DroRI data in MIST is listed without attribution. According to the authors, about half of these are from FlyBi and the other half are from Lit-BM. However, the source database is just listed as “extend” and many lack any PubMedID, even many that are clearly from the literature.

We appreciate the suggestion. We added information about sources to the DroRI tab at MIST.

R2-16) Clearly, the MIST people should be working to integrate the DroRI data with the rest of the Drosophila data in MIST. For example, rather than giving DroRI it’s own tab, the data should be in with the rest of the fly PPI, but with annotations indicating that it is part of DroRI, and found by FlyBi where appropriate. In the meantime, data in DroRI tab should be fully annotated.

As suggested, we have now added the new FlyBi pairs to the Drosophila tab at MIST. We note that our quality standards for inclusion in the Drosophila Reference Interactome (DroRI) as defined here and our quality standards for inclusion in the full Drosophila MIST database are not identical. Thus, we feel there is a benefit to maintaining a dedicated tab for DroRI in addition to a tab for access to the full Drosophila tab at MIST. In addition, as noted above, we have now added information about the source databases to the DroRI tab.

R2-17) A related issue is that MIST site does not explain how the “Ranks” are determined. I found at least one PPI that is ranked “low” in the fly dataset, but high in the DroRI.

The ranking criteria for MIST was described in the relevant publication (PubMed Central ID PMC5753374). As stated in the publication, “A rank of ‘high’ is assigned for interactions supported by multiple experimental methods and/or reported in multiple publications. A rank of ‘moderate’ is assigned if these criteria are not met but the interaction is supported by data from another species.” DroRI is focused on direct binary interactions. As such, in the DroRI set, evidence from the mass-spec experiments is not weighted the same as evidence from a Y2H experiment (indeed, this is one of the reasons we implemented a dedicated tab for DroRI, separate from the Drosophila tab). Some of the FlyBi data submitted into IntAct has been

reannotated by FlyBase. MIST integrates PPI data from public resources. Thus, the FlyBi data from our early submission to IntAct are available in the Drosophila tab by MIST. MIST requires two pieces of evidence in order to define an interaction as high confidence. The FlyBi data in the early submission that does not overlap with any other dataset is assigned a rank of low confidence by MIST. As suggested above, we added the new FlyBi pairs to the Drosophila tab at MIST. For these, we assigned the rank in accordance with the MIST standard.

Reviewer #3 (Remarks to the Author):

The manuscript by Tang et al presents a new and greatly expanded binary interactome for Drosophila, generated by the yeast two-hybrid system supplemented with computationally predicted (and then validated) interactions. I fully agree with the authors that binary interactomes are highly valuable for the scientific community, and that improvements in approaches and analyses make it worthwhile to continue to invest in interactomes for widely-used model systems, like Drosophila.

The experiments have been done well by absolute experts in the field. The experimental and computational pipeline has been developed over the past two decades, and was recently applied to generate a human interactome map. Here, the same pipeline is applied to Drosophila proteins. Starting from validated ORF clones, a highly optimized two-hybrid framework is used that is followed up by quality validation using the orthogonal MAPPIT assay, and computational comparisons with other datasets. The resulting network should be of the highest quality that is currently attainable, and this is evidenced by comparison with high-quality literature derived data. FlyBi will undoubtedly be used by many Drosophila researchers and I highly support its publication. The manuscript ends with an illustrative example where the network is used to gain new insight into autophagy, ultimately focusing on the regulator deformed wings.

I have no experimental suggestions but the clarity of the manuscript could be improved in certain sections.

R3-1) I found the logic of the writing in the section lines 138-150 confusing. It almost seems like the order is not correct or a section of text is missing. In line 137 the description of the pipeline up to and including the retesting and L3 additions are described. I would expect this to be followed by the text starting on line 151 (the numbers that were identified in the pipeline). Instead, on line 138 the authors describe some sort of initial quality test. This starts by establishing 4 sets of protein pairs, but then no quality assessment data is presented for these 4 sets. Instead, the authors describe pairwise testing results for the L3 predictions, without mentioning what kind of pairwise test was performed. These data are only described in text, with no accompanying figure. I also don't understand how 71% of 1000 predictions tested positive, yet only 332 computationally predicted pairs are considered confirmed and added to FlyBi.

The above points were addressed by significant changes to the section noted as confusing. Please see our revised main text section "Expansion of the binary protein-protein interaction network for Drosophila" and our responses to comments R1-1 and R2-6.

Finally, how was the PRS set established? Was this a random selection of FlyBi (that would be ideal for a quality control dataset), or a manually selected set of pairs? Can the authors clarify this section?

The positive reference set (PRS) was curated from the literature. We first applied the same rules we use to define Lit-BM lists (i.e., at least two pieces of evidence for the interaction), then filtered to include only pairs for which both ORFs are represented in the FlyBi ORF collection.

The section of the manuscript that first describes the PRS and RRS now reads as follows: "... we established (i) a small, high-confidence positive reference set (PRS) based on the literature and filtered to include only pairs for which both ORFs are present in the FlyBi ORF collection (Suppl. Table 1), (ii) a random reference set (RRS) of the same size and with the same filter applied (Suppl. Table 1), ..."

R3-2) On line 206 the authors conclude that the level of enrichment in GO and/or phenotype annotations is comparable between FlyBi and Lit-BM-20. Yet in Figure 2C the enrichment odds values for Lit-BM-20 are higher than for FlyBi, particularly since a log scale was used. Why is the conclusion that the levels of enrichment are similar between the two datasets? Can the authors apply statistics to prove this?

We appreciate having this inconsistency brought to our attention. We deleted the statement that originally appeared on line 206. Both sets are significant but as the reviewer rightly points out, they are not comparable (i.e., not similar). The odds values for the Lit-BM are higher.

R3-3) I was surprised by the low overlap between FlyBi and DPiM. I know many arguments why this may be the case, and the authors briefly mention some. Yet I was expecting that as datasets become more complete and more accurate, the overlap would start to increase. Would it be possible to expand the discussion beyond 'experimental and biological differences'? For instance, within a protein complex binary interactions will occur. Not all of these will be able to form between the two proteins in isolation, but intuitively I expect that some fraction will. Is there any literature or analysis that might support that the majority of protein interactions that take place within a protein complex cannot be reconstituted between the two proteins in isolation? Or is there evidence that we have still only scratched the surface of interactions and we would not expect to find overlap?

We thank the reviewer for raising an important discussion point. We share your interest in exploring this seemingly small overlap. We re-did the analysis and found that 72 protein interactions are in common to both FlyBi and DPiM datasets. This represents 2% and 0.9% of interactions detected by Y2H and AP-MS methods, respectively (within the space of protein pairs tested by both assay types). Another study that used both AP-MS and Y2H (in which each interaction was verified and the overall dataset was validated using orthogonal assays), only 6 interactions (representing 1.3% of the interactions detected by Y2H, and 0.16% of interactions detected by TAP-MS) were detected by both methods (Rozenblatt-Rosen, et al. 2012, PMID: 22810586). Moreover, the observed overlap between FlyBi and DPiM datasets is greater than

expected if the assays were completely independent (Fisher's exact test, $P < 2.2e-16$, OR = 45.1).

We updated the main text to reflect the above (paragraph begins, "We also compared the FlyBi network with interaction data from ..."); added a description of the analysis in the Materials and Methods section (new sub-section "Comparison of FlyBi and DPiM" in the "Bioinformatics analysis" section); and added a new Suppl. Table 7.

Furthermore, to understand the fraction of true interactions that our map represents ('completeness'), we also used the empirical framework established for evaluating the quality of interaction maps (Venkatesan et al. 2009, PMID: 19060904; Braun et al. 2009, PMID: 19060903), showing that, as the reviewer has suggested, the vast majority of the interactome remains to be mapped.

More specifically, in the framework of Venkatesan *et al.* 2009, assay sensitivity (S_a) is defined as the fraction of true interactions that can be detected by a given assay. Sampling sensitivity (S_s) is the fraction of detectable true interactions that can be recovered in a given screen. The overall sensitivity S of a given screen ('screening sensitivity') can then be calculated as $S = S_a \times S_s$. In pairwise testing, it is generally assumed that $S_s = 1$, so that the assay sensitivity can be established from the fraction of Positive Reference Set (hsPRS-v1) pairs (Braun et al. 2009; Luck et al 2020 PMID: 32296183) that score positive in the assay. In the absence of benchmarking studies in *Drosophila* interactome and given that for the same assay PRS recovery rates are remarkably similar across interactomes as divergent as yeast (Yu et al. 2008, PMID 18719252) and humans (Luck et al. 2020), we modeled our sensitivity estimates based on the recent benchmarking of Version 1 and Version 3 assays in human interactome (Luck et al. 2020).

Version 1 demonstrated an assay sensitivity of $S_{a-v1} = 21.4\%$, recovering 18 out of 84 PRS pairs (Luck et al. 2020). Sampling sensitivity of Y2H after 2 repeats in 2 orientations has been shown to be $S_{s-v1} = \sim 60\%$ (Altmann et al. 2018, PMID 29944780), allowing us to estimate screening sensitivity as $S_{v1} = S_{a-v1} \times S_{s-v1} = 0.214 \times 0.6 = 12.8\%$. We screened all pairwise combinations of 9,473 out of 13,969 annotated protein-coding genes, corresponding to a search space completeness of 46% ($T_{v1} = 46\%$). Therefore, we can estimate the overall completeness of our Version 1 map to be $C_{v1} = T_{v1} \times S_{v1} = 0.46 \times 0.128 = 5.8\%$.

Version 3 was benchmarked to have an assay sensitivity of $S_{a-v3} = 27.4\%$, recovering 23 out of 84 PRS pairs (Luck et al. 2020). Extrapolating from Version 1 empirical measurements, the screening sensitivity of Version 3 can be estimated to be $S_{v3} = S_{a-v3} \times S_{s-v1} = 0.274 \times 0.6 = 16.4\%$. Because the exact same search space was queried, the overall completeness of Version 3 can be estimated to be $C_{v3} = T_{v3} \times S_{v3} = 0.46 \times 0.165 = 7.6\%$. Only 13 out of 28 (46.4%) hsPRS-v1 pairs detected by the union of Version 1 and Version 3 were detected with both methods, indicating different assay sensitivity profiles of the employed methods and corresponding to the 'orthogonality' of $O_{v1+v2} = 53.6\%$ (i.e. $\sim 50\%$ of *detectable* interactions are different).

Thus, we estimate that the fraction of all true interactions captured by our merged interactome maps is $C_{v1+v3} = (C_{v1} + C_{v3}) \times O_{v1+v3} \cong (0.058 + 0.076) \times 0.536 = 7.2\%$. Together, Version 1 and Version 3 screens yielded 8714 interactions (excluding computational predictions), so we can estimate the total *Drosophila melanogaster* interactome size to be $\sim 120K$ interactions, which is on par with a previous estimate (Stumpf et al. 2008, PMID: 18474861). In summary, we estimate that our FlyBi interactome represents $\sim 7\%$ of the complete binary interactome.

Quantitative estimation of the expected overlap is made difficult by the fact that AP-MS often captures indirect associations whereas Y2H tends to find only direct physical interactions, and more so because AP-MS has never been subjected to the same empirical framework to estimate precision and sensitivity. Although systematic benchmarking data for AP-MS is lacking, we modeled the sensitivity of AP-MS screens (S_{AP-MS}) as falling between 10% and 80% (we expect that 80% is a conservatively high estimate given that all other systematically benchmarked assays exhibited a sensitivity between 10% and 35% (3,8)). We can further model from Luck *et al.* 2020 that 30% of the interactions detected by AP-MS are direct ($D_{AP-MS} = 0.3$). Then, the expected fraction of AP-MS interactions recovered by Y2H would be $E = S_{Y2H} \times D_{AP-MS} \times S_{AP-MS} = 0.15 \times 0.3 \times [0.1, 0.8] = 0.45 - 3.6\%$.

Thus, the 72 overlapping interactions we observed are within the range of what might have been expected ($[0.0045, 0.036] \times 8,205 = 37 - 295$ interactions).

Although we believe the analysis presented in detail above is valid, e.g., that it is valid to use sampling sensitivity from human Y2H screens because the same assay versions were applied, we erred on the side of caution and did not add this analysis to the manuscript.

R3-4) Quite a high fraction of genes tested modify the autophagy eye phenotype of Atg1 overexpressing flies. The authors nicely perform a control experiment to show that this is not (or to a lesser extent) the case for a random selection of genes. In the larval Fat body, again a large fraction of genes modified the Atg8a puncta formation (positively or negatively). A comparison between assay (overlap in results) would be informative.

After identifying candidate genes that modified the Atg1-induced rough eye phenotype, we further verified these putative autophagy regulators (i.e., the candidate genes for which RNAi knockdown resulted in rescue/enhancement of Atg1-induced eye phenotypes). We used Atg8a puncta formation in the larval fat body as a readout to test their roles in autophagy. Because these candidates were selected based on their performance in the eye screen, it is not surprising that a large fraction were able to affect Atg8a puncta formation. The eye and fat body screen results are listed in the Suppl. Table 8 (previous Suppl. Table 10). Overall, of the 234 RNAi lines identified in the eye screen, 60 (26%) increased fat body Atg8a puncta under fed conditions, and 41 (18%) inhibited fat body Atg8a puncta formation upon starvation.

R3-5) Line 300: a 55% retest rate by IP is in line with previous retest rates of high quality datasets I believe. I would be useful to indicate if this is the case.

We appreciate the comment and investigated the literature. We did indeed find studies that report similar rates. In a recent paper published by Alcantara *et al.*, 5 out of 12 (42%) interactions from Y2H can be confirmed by immunoprecipitation (Alcantara *et al.*, 2019). In another study, 45 out of 79 (57%) putative interactions identified from Y2H screens were positive in CoIP experiments (To *et al.*, 2011). These results suggest that the positive rates of IP are in line with high quality of Y2H dataset. We added a brief note about this in the main text (immediately preceding the subsection “Dwg suppresses autophagy by binding to insulator elements near ATG genes”).

R3-6) Line 507: are positive pairs picked into individual wells? It wasn't clear to me if the PCR analysis in this step does not have to deal with the potential of multiple interactions in one well (I assume so)

Yes, multiple interactions per well is a possibility. According to our well-established workflow, we handled things as follows to address this. Colonies from the screens were picked into individual wells. Yeast colonies from a given spot can theoretically contain distinct AD interactors (we tested one bait against 1000 preys). If several yeast colonies grew on one spot, we picked up to 5 colonies per spot into individual wells. We then performed SWIM-seq on all picked colonies (see "Yeast colony sequencing" in the Methods section). Based on the SWIM-score, we are able to assign which proteins interacted in that well. We then performed pairwise tests with all of these first-pass pairs (FiPPs). If an interaction occurs in a pairwise test and the yeast cells grow on the 3AT selection media and not on media containing CHX (auto-activator test), we pick only one colony of the yeast spot and then do another round of SWIM-seq (where only one bait and prey are being amplified with barcodes in each well but the 96-well PCR is still being pooled). Based on the barcodes, which are added during the PCR amplification and are different in each well for each AD and DB, we can confirm that the right pair was tested and that it's a true interaction.

R3-7) Line 509 refers to Luck et al for computational analysis to generate a list of binary interactors identified in the screens. It wasn't clear to me what analysis this refers to. In the retest section of the referred paper the most relevant I could find was 'Only pairs identified from the pipeline that matched the tested ORF pairs previously scored as positive were considered PPIs.', but that is likely not what is meant by computational analysis.

By "computational analysis" here we meant the SWIM score and how it's calculated, which we also describe in the "Yeast colony sequencing" sub-section of the Methods. Using the SWIM score, we selected a set of FiPPs identified in the screens, which we then pairwise tested. We updated the manuscript text to replace "computational analysis" with "SWIM score" for clarity.

R3-8) Line 136 refers to Fig. 1A to support the supplementing of the network with validated computational predictions, but there is no data/drawing regarding computational predictions in this panel.

We apologize for the error. We removed the reference to Fig. 1A from the relevant section.

R3-9) Line 137: experimentally predictions should presumably read experimentally validated computational predictions

We corrected this mistake as suggested (the line now reads "experimentally validated").

REVIEWERS' COMMENTS

Reviewer #1 (Remarks to the Author):

The authors well address comments raised from the review of the initial submission (which were rather minor). I suggest to go ahead with the publication of this new drosophila interactome data.

Reviewer #2 (Remarks to the Author):

The revised manuscript is excellent and adequately addresses all of this reviewer's concerns about the previous version.

Here are minor comments for the authors to consider.

The description of the computational approach to predicting PPI is much improved but it still does not explain how the approach landed on "332 experimentally validated computational predictions". It was noted that the experimental validation rates for the top 100 predictions was ~90% (so, 90?), and for the top 500 was ~80% (so, 400?) and so on, but how do any of those numbers add up to 332?

Supp table 6 is a good summary of the tested interactions. It would be useful to also include the reference column that was added to other tables. Doing so will reveal that there is overlap between the different sets that were tested. Some of the MAPPIT tested interactions came from more than one dataset, even though table 6 and Supp Fig 3 assign them to just one dataset.

Also, since these were all random selections from each dataset, it may be more appropriate to include results for all of the MAPPIT tested PPI from each dataset, rather than just the PPI that were initially assigned or selected directly from that dataset. This would change the results (to be more accurate, in my view), though probably not the overall trends and conclusions.

Reviewer #3 (Remarks to the Author):

Overall, the clarity of the manuscript is much improved, and the points I and other reviewers raised appear well addressed. I enjoyed reading the extensive answer to my question regarding overlap between Y2H and AP/MS, and appreciate that it is beyond the scope of this manuscript to include. I strongly support publishing of this important work.

I have only one remaining issue. While the text describing the computational prediction of interactions is much clearer, I still do not understand how the additional 332 interactions were arrived at. The methods section (line 539) states that 'we experimentally tested the top 1,000 predictions from the L3 computational analysis'. With a retest rate of 71% (line 152), to me it reads as if that should have resulted in the addition of 710 additional interactions rather than 332. Maybe I'm just missing something but it would be great if the authors can clarify this.

Responses to Reviewers' Comments

Reviewer #1 (Remarks to the Author):

The authors well address comments raised from the review of the initial submission (which were rather minor). I suggest to go ahead with the publication of this new drosophila interactome data.

Response: We thank the reviewer for the supportive response.

Reviewer #2 (Remarks to the Author):

The revised manuscript is excellent and adequately addresses all of this reviewer's concerns about the previous version.

Here are minor comments for the authors to consider.

The description of the computational approach to predicting PPI is much improved but it still does not explain how the approach landed on "332 experimentally validated computational predictions". It was noted that the experimental validation rates for the top 100 predictions was ~90% (so, 90?), and for the top 500 was ~80% (so, 400?) and so on, but how do any of those numbers add up to 332?

We appreciate the comment and apologize for the continued confusion about the specifics regarding this section of the study. We have further updated the main text to provide more clarity.

The revised passage in the Results section is as follows (new text is underlined): "The list of sequence-confirmed putative binary interactors resulting from experimental Y2H testing was supplemented ... Following application of the L3 approach to data from the screens 1 and 2, we experimentally tested the top 1,000 computationally predicted pairs. Excluding 254 undetermined pairs (see Methods), 71%, 80% and 90% of the top 1000, 500 and 100 predictions were scored as positives, comparing to 13% for Lit-BM-16 pairs. Of the 533 positive predictions, 332 interactions were sequence-confirmed and added to the FlyBi dataset."

The following passage was added to the "Pairwise test of FiPPs" section of the Methods (new text is underlined): "Each FiPP was subjected ... final FlyBi dataset. A protein pair was scored as positive only when significantly more growth was observed on the test plate compared to the CHX plate. In the case of too strong growth on CHX plate, a pair was scored as auto-activator (classified as undetermined). If there was no growth on test and CHX plate, the pair was scored as negative. A pair was scored undetermined (NA) if the well was unscorable (contaminated, not spotted, etc.)."

As further clarification, we offer the following:

As noted in this section of the Results, we used experimental pairwise testing to assess the quality of the top 1,000 computationally predicted pairs. A protein pair was scored as positive

only when significantly more growth was observed on the test plate compared to the CHX plate. In the case of too strong growth on CHX plate, a pair was scored as an auto-activator (classified as undetermined). If there was no yeast spotted on the plate, the pair was scored as invalid (classified as undetermined). If there was no growth on test and CHX plate, the pair was scored as negative. The 1000 pairs resulted in 533 positives and 213 negatives, along with 254 pairs that were undetermined and thus removed from calculations and further study. We calculated precision (percent positive as reported for top 1000, top 500, etc.) following the conventional definition: positives / (positives + negatives). For predicted pairs scoring among the top 1,000 predictions, this value was $533 / (533 + 213) = 71.4\%$. For predicted pairs in the top 500, this value is $80 \pm 5\%$. For predicted pairs in the top 100, it was $90 \pm 10\%$. Pairs curated from the literature were recovered at lower rates ($13 \pm 3\%$ for Lit-BM-16 pairs). The values observed for pairs predicted using the L3 approach indicate strong, highly non-random network patterns. To apply added rigor to pairs to be included in the FlyBi dataset, we next confirmed the predicted pairs that had scored as positive pairs using sequence confirmation. A subset of 332 pairs were confirmed by sequencing. These pairs were added to the FlyBi dataset.

Supp table 6 is a good summary of the tested interactions. It would be useful to also include the reference column that was added to other tables. Doing so will reveal that there is overlap between the different sets that were tested. Some of the MAPPIT tested interactions came from more than one dataset, even though table 6 and Supp Fig 3 assign them to just one dataset.

Response: Regarding the first part of the comment, we appreciate the suggestion and added information to the table as suggested (the updated table is included in the current version as Supplementary File 6). Regarding the second part of the comment, we recognize that there are multiple ways to present data and feel that our presentation is reasonable and sufficient.

Reviewer #3 (Remarks to the Author):

Overall, the clarity of the manuscript is much improved, and the points I and other reviewers raised appear well addressed. I enjoyed reading the extensive answer to my question regarding overlap between Y2H and AP/MS, and appreciate that it is beyond the scope of this manuscript to include. I strongly support publishing of this important work.

I have only one remaining issue. While the text describing the computational prediction of interactions is much clearer, I still do not understand how the additional 332 interactions were arrived at. The methods section (line 539) states that ‘we experimentally tested the top 1,000 predictions from the L3 computational analysis’. With a retest rate of 71% (line 152), to me it reads as if that should have resulted in the addition of 710 additional interactions rather than 332. Maybe I’m just missing something but it would be great if the authors can clarify this.

We appreciate this concern, which was shared by another reviewer. We updated the main text to provide more clarity. Please see above.